# Ultrafast electron transfer at the $In_2O_3$/$Nb_2O_5$ S-scheme interface for $CO_2$ photoreduction

Xianyu Deng[1,4], Jianjun Zhang [1,4], Kezhen Qi[2], Guijie Liang [3], Feiyan Xu [1] ✉ & Jiaguo Yu [1] ✉

Constructing S-scheme heterojunctions proves proficient in achieving the spatial separation of potent photogenerated charge carriers for their participation in photoreactions. Nonetheless, the restricted contact areas between two phases within S-scheme heterostructures lead to inefficient interfacial charge transport, resulting in low photocatalytic efficiency from a kinetic perspective. Here, $In_2O_3$/$Nb_2O_5$ S-scheme heterojunctions are fabricated through a straightforward one-step electrospinning technique, enabling intimate contact between the two phases and thereby fostering ultrafast interfacial electron transfer (<10 ps), as analyzed via femtosecond transient absorption spectroscopy. As a result, powerful photo-electrons and holes accumulate in the $Nb_2O_5$ conduction band and $In_2O_3$ valence band, respectively, exhibiting extended long lifetimes and facilitating their involvement in subsequent photoreactions. Combined with the efficient chemisorption and activation of stable $CO_2$ on the $Nb_2O_5$, the resulting $In_2O_3$/$Nb_2O_5$ hybrid nanofibers demonstrate improved photocatalytic performance for $CO_2$ conversion.

Excessive emissions of carbon dioxide ($CO_2$) into the atmosphere have disrupted the natural carbon cycle, leading to severe environmental consequences, particularly the exacerbation of the greenhouse effect[1-6]. In response to this urgent global issue, harnessing abundant, clean, and inexhaustible sunlight to convert $CO_2$ into valuable solar fuels has emerged as a promising strategy[7-13]. However, the effectiveness of $CO_2$ photoreduction is constrained by the challenging chemisorption and activation of $CO_2$ molecules on catalysts, primarily due to the high dissociation energy of the C=O bond (-750 kJ mol$^{-1}$)[14-19]. Therefore, the development of advanced photocatalysts proficient in activating $CO_2$ has become a pivotal concern within the realm of photocatalytic $CO_2$ reduction[20-26]. Niobium pentoxide ($Nb_2O_5$), a non-toxic solid oxide renowned for its high conduction band (CB) level and potent reduction capability, has recently gained significant attention in photocatalysis[27-33]. Preliminary density functional theory (DFT) calculations indicate that $CO_2$ molecules adsorbed onto the $Nb_2O_5$ undergo changes in both bond lengths and angle compared to free ones.

Additionally, the two oxygen atoms of $CO_2$ can form chemical bonds with niobium atoms of $Nb_2O_5$, suggesting the potential of $Nb_2O_5$ for activating stable $CO_2$ molecules during $CO_2$ photoreduction. Nevertheless, unitary $Nb_2O_5$ exhibits poor photocatalytic performance resulting from sluggish electron/hole separation and charge transfer kinetics. Consequently, developing hybrid heterojunctions involving $Nb_2O_5$, capable of activating $CO_2$, promoting charge carrier transfer kinetics, and separating them to improve reduction efficiency, remains a significant yet challenging endeavor.

S-scheme heterojunctions, integrating both reduction and oxidation photocatalysts, have proven effective in spatially separating photogenerated charge carriers with robust redox capabilities[34-39]. Conventionally, constructing S-scheme heterojunctions involves initially acquiring photocatalyst I and subsequently applying photocatalyst II onto I through methods such as in situ growth or electrostatic self-assembly[40-42]. However, these post-hybridization methods cannot ensure the maximum contact area between the two phases at atomic

[1]Laboratory of Solar Fuel, Faculty of Materials Science and Chemistry, China University of Geosciences, Wuhan 430078, PR China. [2]College of Pharmacy, Dali University, Dali 671003, PR China. [3]Hubei Key Laboratory of Low Dimensional Optoelectronic Materials and Devices, Hubei University of Arts and Science, Xiangyang 441053, PR China. [4]These authors contributed equally: Xianyu Deng, Jianjun Zhang. ✉e-mail: xufeiyan@cug.edu.cn; yujiaguo93@cug.edu.cn

levels, thus impeding the efficient interfacial transport of photo-generated carriers and compromising photocatalytic efficiency (Fig. 1a)[43]. In this study, we designed an S-scheme heterojunction by coupling $Nb_2O_5$ with indium oxide ($In_2O_3$), an oxidation photocatalyst with a narrow bandgap (~2.9 eV) and visible light absorption[44-51]. By mixing the precursors of both phases in the same electrospinning solution, $In_2O_3$ and $Nb_2O_5$ are simultaneously formed during the high-temperature calcination of the electrospun nanofibers. This "one-pot" preparation method ensures maximum phase contact without any hindrance, providing an unimpeded transport route and promoting interfacial charge transfer between $In_2O_3$ and $Nb_2O_5$ (Fig. 1a). Analysis using femtosecond transient absorption spectroscopy (fs-TAS) revealed ultrafast photoelectron transfer from the $In_2O_3$ CB to the $Nb_2O_5$ valence band (VB), inhibiting self-carrier recombination, effectively segregating powerful photoelectrons in the $Nb_2O_5$ CB and the holes in the $In_2O_3$ VB, as well as prolonging the long-lifetimes of the nanohybrids. Also benefiting from the chemisorption and activation of $CO_2$ molecules on the catalyst, the resulting $In_2O_3/Nb_2O_5$ heterojunctions demonstrated enhanced performance in $CO_2$ photoreduction.

This work provides insights into ultrafast charge transfer at the S-scheme heterojunction interface through fs-TAS investigations, offering an essential understanding for the development of hetero-junctions and broadening their potential applications in artificial photosynthesis.

## Results and discussion

### Characterizations and charge separation mechanism of $In_2O_3/Nb_2O_5$ heterojunctions

The $In_2O_3/Nb_2O_5$ heterojunctions, synthesized through a one-step electrospinning procedure, are designated as IN$x$, where I and N represent $In_2O_3$ and $Nb_2O_5$, respectively, and $x$ signifies the weight percentage of $Nb_2O_5$ relative to $In_2O_3$. The precise $Nb_2O_5$ content of all composites was determined using inductively coupled plasma-atomic emission spectrometry (ICP-AES), and the results are presented in Supplementary Table 1. Field emission scanning electron microscopy (FESEM) images reveal distinctive morphologies: pure $In_2O_3$ exhibits a fibrous structure, while pristine $Nb_2O_5$ displays a clubbed pattern (Supplementary Fig. 1). The $In_2O_3/Nb_2O_5$ heterojunction (IN10) shows a

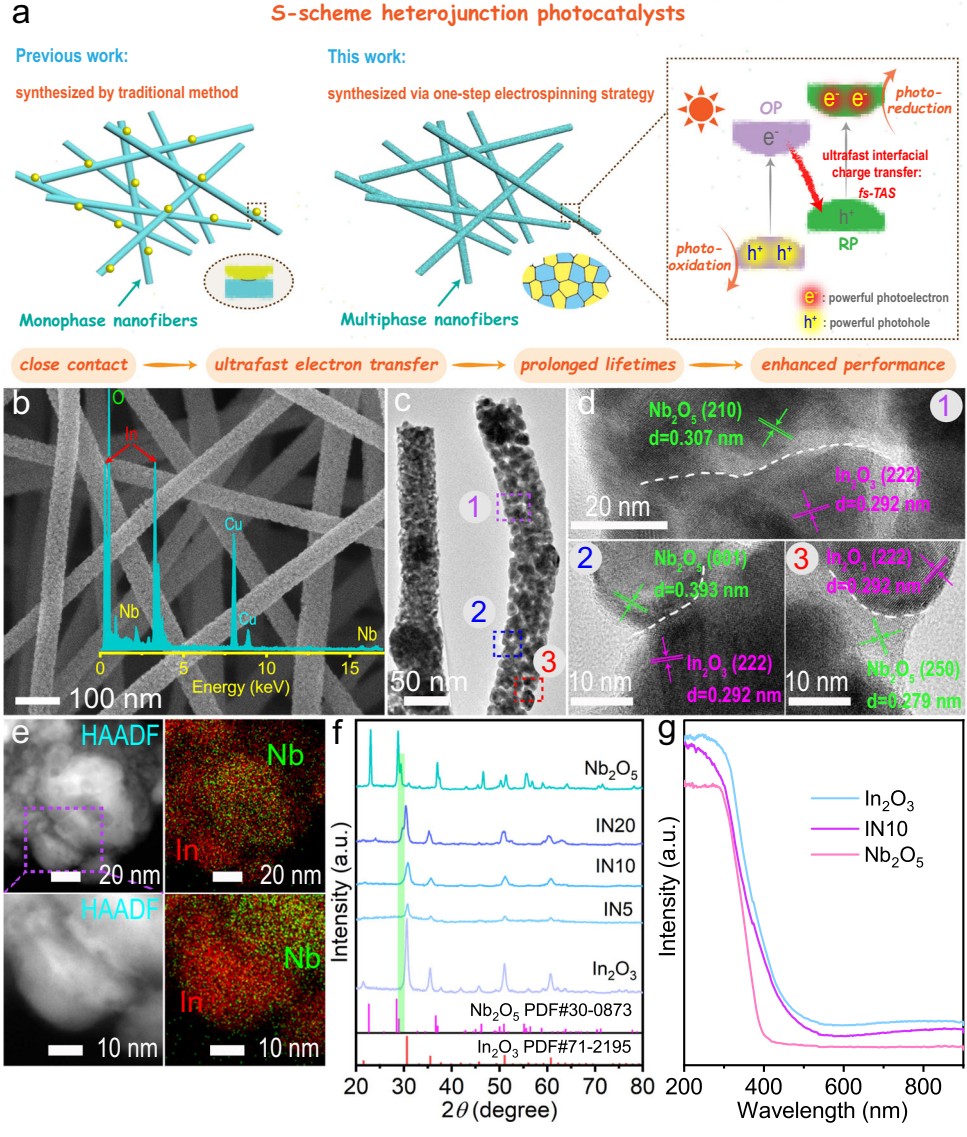

**Fig. 1 | Morphology and structure of $In_2O_3/Nb_2O_5$ heterojunctions. a** Schematic and the design concept of this study. OP and RP stand for oxidation photocatalyst and reduction photocatalyst, respectively. **b** FESEM image and EDX spectrum, (**c**) TEM image, and (**d**) HRTEM images of $In_2O_3/Nb_2O_5$ heterojunctions (IN10). **e** High-angle annular dark-field (HAADF) image and EDX elemental mappings of In and Nb elements in IN10 at different magnifications. **f** XRD patterns of $In_2O_3$, $Nb_2O_5$ and IN$x$. **g** UV-vis spectra of $In_2O_3$, $Nb_2O_5$, and IN10.

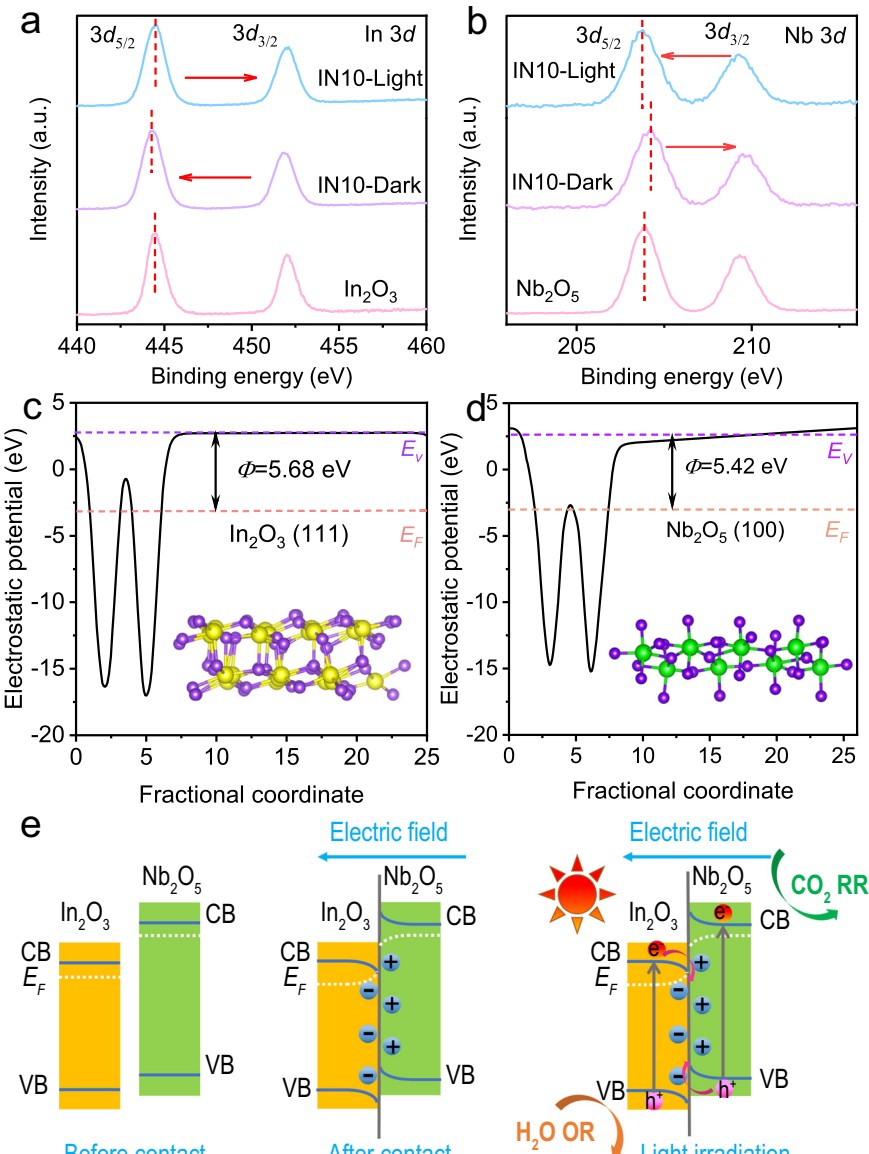

**Fig. 2 | Electron transfer between In₂O₃ and Nb₂O₅ within the heterojunctions.** The high-resolution XPS spectra of (**a**) In $3d$, and (**b**) Nb $3d$ of In₂O₃, Nb₂O₅, and IN10. Calculated electrostatic potentials of (**c**) In₂O₃ (111) and (**d**) Nb₂O₅ (100) slabs. The yellow, green, and purple spheres represent In, Nb, and O atoms, respectively.

**e** The formation of In₂O₃/Nb₂O₅ S-scheme heterojunction, and the proposed charge transfer and separation mechanism. RR and OR stand for reduction reaction and oxidation reaction, respectively.

uniform fibrous morphology with a rough surface and diameters below 100 nm, as observed via both FESEM and transmission electron microscopy (TEM) (Fig. 1b, c). High-resolution TEM (HRTEM) images of IN10 reveal discernible two-phase grain boundaries with lattice fringes corresponding to In₂O₃ and Nb₂O₅, respectively (Fig. 1d). The random distribution of In₂O₃ and Nb₂O₅ nanoparticles within the nanofibers ensures close contact, facilitating unimpeded interfacial transport and efficient separation of photoexcited charge carriers (as discussed below). Energy-dispersive X-ray (EDX) analysis (inset in Fig. 1b) and elemental mappings of IN10 (Supplementary Fig. 2) unambiguously confirm the presence of In, Nb, and O elements within the nanohybrid, offering compelling evidence for the formation of In₂O₃/Nb₂O₅ heterojunctions. Figure 1e displays enlarged elemental mappings targeting a grain-to-grain area within IN10. It is apparent that the distribution of Nb and In elements exhibits a non-overlapping pattern along the interface of the two phases. X-ray diffraction (XRD) patterns of pure In₂O₃ nanofibers (Fig. 1f) indicate the monoclinic phase (PDF#71-2195), while characteristic peaks corresponding to Nb₂O₅ (PDF#30-0873)

emerge in the IN*x* composites at 20 wt.% Nb₂O₅ content, signifying the successful synthesis of In₂O₃/Nb₂O₅ heterojunctions. UV-vis diffuse reflectance spectroscopy (DRS) delineates the optical properties of In₂O₃, Nb₂O₅, and In₂O₃/Nb₂O₅ composites (Fig. 1g). The absorption edges of pristine In₂O₃ and Nb₂O₅ are positioned at 425 and 390 nm, corresponding to bandgaps of 2.9 and 3.2 eV, respectively. Compared to pure Nb₂O₅, the slightly improved UV and visible light absorption characteristics of IN10 suggest successful hybridization due to the strong light absorption capacity of In₂O₃.

X-ray photoelectron spectroscopy (XPS) was utilized to analyze the surface chemical states and compositions of the resulting samples. The survey spectrum of IN10 reveals the presence of In, Nb, and O elements in the hybrid nanofibers (Supplementary Fig. 3). In the high-resolution In $3d$ XPS spectra (Fig. 2a), two distinct peaks are observed at 444.5 and 452.0 eV, corresponding to the $3d_{5/2}$ and $3d_{3/2}$ states of trivalent In³⁺ within In₂O₃, respectively. The signals associated with Nb $3d_{5/2}$ and Nb $3d_{3/2}$ appear at 206.9 and 209.7 eV, respectively, confirming the existence of pentavalent Nb⁵⁺ in the samples (Fig. 2b)[30]. The

O 1$s$ XPS spectra of $In_2O_3$, IN10, and $Nb_2O_5$ (Supplementary Fig. 4a–c) consistently exhibit peaks attributed to lattice oxygen and surface hydroxyls (-OH). Notably, in the IN10 composite, the binding energies (BEs) of In 3$d$ show negative shifts compared to pure $In_2O_3$, while the peaks of Nb 3$d$ shift towards higher BEs in comparison with bare $Nb_2O_5$. These shifts suggest that electrons are transferred from $Nb_2O_5$ to $In_2O_3$ upon contact, indicating the creation of a directional inter-facial electric field (IEF) from $Nb_2O_5$ to $In_2O_3$, and simultaneously leading to the bending of the energy bands at the interfaces. To further substantiate the electron transfer process between $In_2O_3$ and $Nb_2O_5$, the work function ($\Phi$) was determined through DFT simulations by calculating the energy difference between the vacuum and Fermi levels, based on the electrostatic potential of the materials. As illustrated in Fig. 2c, d and Supplementary Figs. 5, 6, the estimated $\Phi$ value of $In_2O_3$ (111) is larger than that of $Nb_2O_5$ (100), with both facets exhibiting the lowest surface energy (Supplementary Tables 2 and 3). Consequently, $Nb_2O_5$ possesses a higher Fermi level ($E_F$) than $In_2O_3$, promoting the transfer of electrons from $Nb_2O_5$ to $In_2O_3$ until reaching the same $E_F$ at the interface (Fig. 2e). These analyses align with the aforementioned XPS results and contribute to the efficient separation of photogenerated charge carriers (as discussed below).

To investigate the photoinduced charge transfer mechanism of the $In_2O_3$/$Nb_2O_5$ heterojunctions, the band structure of $In_2O_3$ and $Nb_2O_5$ was first studied. According to the ultraviolet photoelectron spectroscopy (UPS) spectra (Supplementary Fig. 7a, b), the VB maximum of $In_2O_3$ and $Nb_2O_5$ is estimated at 2.41 and 2.28 V (vs. standard hydrogen electrode, SHE), respectively. Combined with the bandgap values disclosed in Supplementary Fig. 7c, the CB minimum is established as −0.49 V for $In_2O_3$ and −0.96 V for $Nb_2O_5$ (Supplementary Fig. 7d)[33]. Based on the previous discussion involving XPS and DFT results, it is evident that the $E_F$ of $Nb_2O_5$ is higher than that of $In_2O_3$, which induces the migration of electrons from $Nb_2O_5$ to $In_2O_3$, leading to the creation of an IEF at the interface, as well as band alignment upon contact. Under light irradiation, electrons in the $In_2O_3$ and $Nb_2O_5$ VBs are initially excited to their respective CBs. Due to the bent energy bands, the IEF with the direction from $Nb_2O_5$ to $In_2O_3$, and the Coulomb attraction between electrons and holes, the photogenerated electrons in the $In_2O_3$ CB tend to transfer to the $Nb_2O_5$ VB and recombine with its holes. These consumed photoelectrons and photoholes are characterized by their weak reduction and oxidation capacities. As a result, photogenerated charge carriers with strong redox capabilities within the $Nb_2O_5$ CB and the $In_2O_3$ VB undergo separation and preservation, actively participating in subsequent photoreactions. This charge transfer pathway implies the formation of an S-scheme heterojunction between $In_2O_3$ and $Nb_2O_5$, visually depicted in Fig. 2e.

In situ irradiated XPS was conducted to verify the S-scheme charge transfer route within the $In_2O_3$/$Nb_2O_5$ heterojunctions. As shown in Fig. 2a, b, upon exposure to light, the BEs of In 3$d$ in IN10 display notable positive shifts, while the Nb 3$d$ peaks shift towards lower BEs, with respect to those in the dark. These observed BE shifts provide strong evidence for the transfer of photogenerated electrons from $In_2O_3$ to $Nb_2O_5$, thus corroborating the proposed S-scheme photocatalytic mechanism. The efficiency of charge separation in the $In_2O_3$/$Nb_2O_5$ S-scheme heterojunctions was assessed through steady-state photoluminescence (PL) and photoelectrochemical measurements. The PL emission intensity of the IN10 composite is weaker than both pure $In_2O_3$ and $Nb_2O_5$ (Supplementary Fig. 8), indicating a significant inhibition of the electron/hole recombination within the $In_2O_3$/$Nb_2O_5$ S-scheme hybrid nanofibers. Moreover, during the long-term photoelectrochemical test, IN10 consistently exhibits the highest and most stable photocurrent density in contrast to pristine $In_2O_3$ and $Nb_2O_5$ (Supplementary Fig. 9), underscoring the efficient charge separation within the $In_2O_3$/$Nb_2O_5$ nanohybrids. Electrochemical impedance spectroscopy (EIS) results demonstrate that IN10 displays a

smaller arc radius in the Nyquist plot compared to bare $In_2O_3$ and $Nb_2O_5$ (Supplementary Fig. 10), signifying a lower charge transfer resistance in the $In_2O_3$/$Nb_2O_5$ composite. These analyses collectively confirm that the hybridization of $In_2O_3$ and $Nb_2O_5$ to form S-scheme heterojunctions can boost charge transfer and effectively reduce the electron/hole recombination, thus facilitating high-efficiency photo-catalytic $CO_2$ reduction[52–56].

The accumulation of photogenerated electrons and holes after S-scheme charge separation was explored through electron paramagnetic resonance (EPR) spectroscopy. The reduction potential of 5,5-dimethyl-1-pyrroline N-oxide (DMPO)-•$O_2^-$ and the oxidation potential of DMPO-•OH are −0.74 and 2.28 V (vs. SHE), respectively. Compared to pristine $In_2O_3$ or $Nb_2O_5$, the $In_2O_3$/$Nb_2O_5$ composite shows intensive EPR signals for both •$O_2^-$ and •OH radicals (Supplementary Fig. 11). This observation signifies the efficient separation and accumulation of energetic photoelectrons in the $Nb_2O_5$ CB and photoholes in the $In_2O_3$ VB, providing compelling evidence for the S-scheme charge separation mechanism.

## Ultrafast electron transfer at the $In_2O_3$/$Nb_2O_5$ S-scheme heterojunction interface

Fs-TAS was employed to delve deeper into the dynamics of photoelectron transfer at the $In_2O_3$/$Nb_2O_5$ S-scheme interface. As depicted in Fig. 3a–f, both pristine $In_2O_3$ and the $In_2O_3$/$Nb_2O_5$ heterojunctions (IN5 and IN10) exhibit noticeable negative peaks at ~480 nm when excited at 340 nm, which correspond to the ground state bleaching (GSB) signals of $In_2O_3$ and provide insights into the population of photoelectrons in its CB[57]. This assignment was further supported by an experiment using $AgNO_3$ as an electron scavenger, wherein the signal virtually disappears (Supplementary Fig. 12), indicating that the photogenerated electrons are trapped by the scavenger, leaving no electrons to recombine with the holes. The normalized recovery kinetics of pristine $In_2O_3$ at 480 nm, monitored within 50 ps, were fitted with a two-exponential function (Fig. 3g and Supplementary Table 4), assigning to the interband diffusion (Process I) and the trapping by shallow trap states (Process II) of the photogenerated electrons in the $In_2O_3$ CB. Upon integrating $In_2O_3$ with $Nb_2O_5$, an additional ultrafast pathway (<10 ps) emerges for the photoelectrons in the $In_2O_3$ CB, namely, their transfer to the $Nb_2O_5$ VB (Process III)[58]. Notably, in an Ar atmosphere, both $\tau_1$ and $\tau_2$ lifetimes demonstrate a gradual decrease with increasing $Nb_2O_5$ content (IN5 and IN10, Fig. 3h, i and Supplementary Table 4). This implies the rapid migration of more photoelectrons from the $In_2O_3$ CB to $Nb_2O_5$ upon hybridization, resulting in fewer electrons available for diffusion and trapping processes (Fig. 3k, l). Under a $CO_2$ atmosphere (Fig. 3j, m), photoelectrons in the $Nb_2O_5$ CB react with $CO_2$ molecules, accelerating the transfer of more photoelectrons from the $In_2O_3$ CB to the $Nb_2O_5$ VB to recombine with its photoholes, thereby shortening the lifetimes. The fs-TAS analysis of a physically-mixed composite of $In_2O_3$ and $Nb_2O_5$ reveals longer lifetimes ($\tau_1$ and $\tau_2$) compared to $In_2O_3$/$Nb_2O_5$ heterostructures (Supplementary Fig. 13), indicating inefficient electron transfer from $In_2O_3$ to $Nb_2O_5$ and underscoring the advantages of interfacial phase contact within the S-scheme heterojunction for efficient charge transfer and separation.

On the other hand, the broad GSB signal of $Nb_2O_5$ appears around 500 nm (Fig. 4a, b), corresponding to the population of photoholes in its VB as affirmed by the disappearance of the signal after introducing a hole-trapping agent (lactic acid) (Supplementary Fig. 14). To minimize the interference of $In_2O_3$, kinetic decay curves of bare $Nb_2O_5$ and the $In_2O_3$/$Nb_2O_5$ nanohybrids (IN20 and IN10) are derived at 530 nm (Fig. 4c–e), which involve two processes related to photoexcited holes in the $Nb_2O_5$ VB, i.e., recombination with self-generated photoelectrons (process 1) and recombination with photoelectrons transferred from $In_2O_3$ (process 2) (Fig. 4g). Under an Ar atmosphere, the half-life ($\tau_{1/2}$) of both IN20 and IN10 is shorter than that of pristine $Nb_2O_5$,

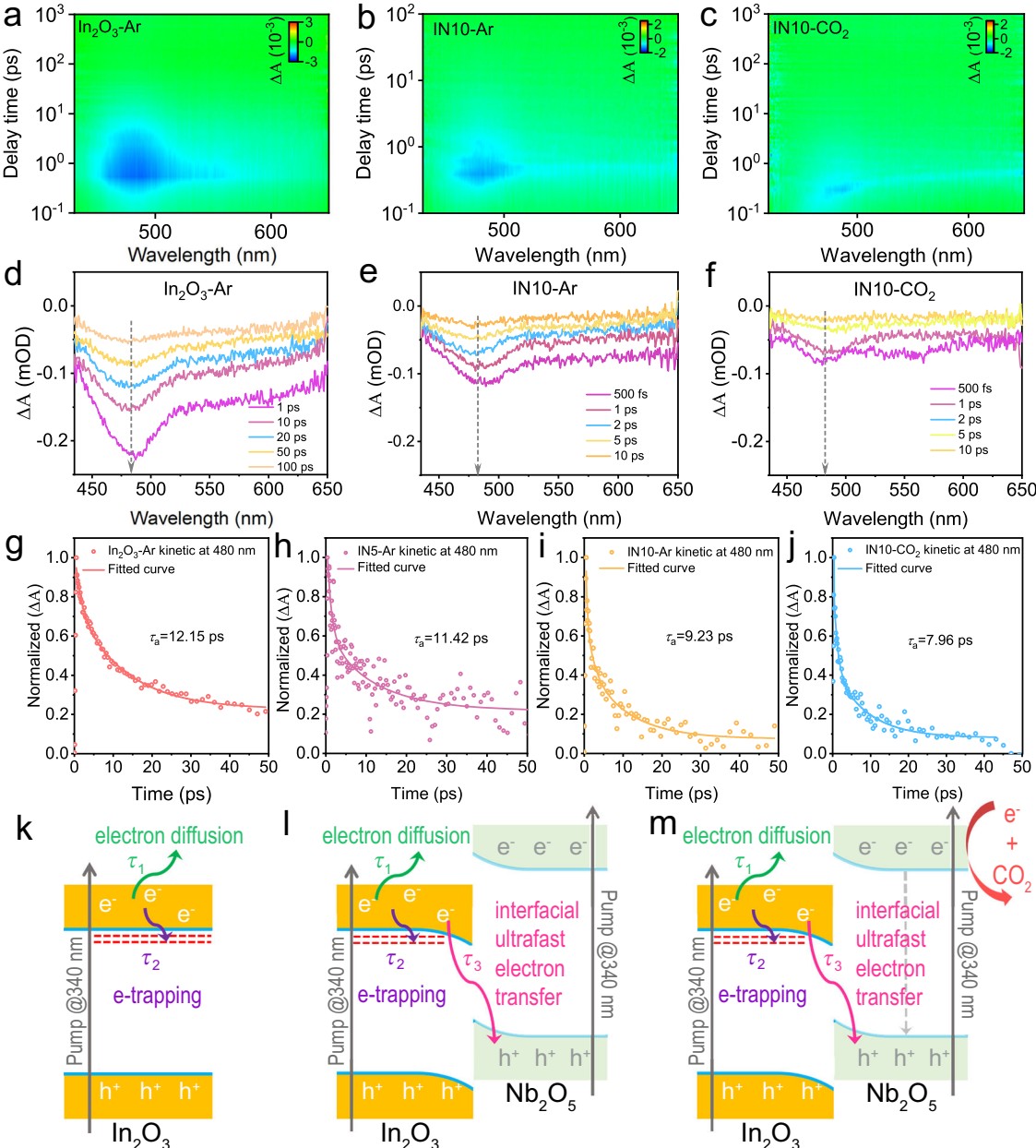

**Fig. 3 | Insights into charge transfer dynamics in pure In₂O₃ and In₂O₃/Nb₂O₅ S-scheme heterojunctions.** The pseudocolor plots and transient absorption spectra recorded at indicated delay times measured with 340 nm excitation: (**a, d**) pure In₂O₃ in Ar, (**b, e**) IN10 in Ar, and (**c, f**) IN10 in CO₂. Corresponding kinetic decay curves at 480 nm within 50 ps: **g** pure In₂O₃ in Ar, (**h**) IN5 in Ar, (**i**) IN10 in Ar, and (**j**) IN10 in CO₂. The decay pathways of photogenerated electrons in (**k**) pure In₂O₃, **l** In₂O₃/Nb₂O₅ heterojunctions in Ar, and (**m**) In₂O₃/Nb₂O₅ heterojunctions in CO₂.

signifying the migration of photoelectrons from the In₂O₃ CB to Nb₂O₅ and thereby reducing the number of VB photoholes (Fig. 4h). In addition, the In₂O₃/Nb₂O₅ heterojunctions exhibit a composition-dependent half-life, with a shorter value as Nb₂O₅ content decreases (IN10 < IN20). According to the S-scheme charge separation mechanism, photoelectrons in the In₂O₃ CB transfer to the Nb₂O₅ VB and recombine with its photoholes in equal proportions. At an ideal In₂O₃/Nb₂O₅ ratio (i.e., IN10), the population of photogenerated charge carriers is approximatively identical in both materials, leaving few excess photoholes in the Nb₂O₅ VB and consequently shortening the lifetime after S-scheme charge separation. When the Nb₂O₅ content deviates from its optimal value (i.e., IN20), some excessive photoholes remain in the Nb₂O₅ VB, leading to a longer lifetime than IN10. Under a CO₂ atmosphere, photoelectrons in the Nb₂O₅ CB actively react with

CO₂, diminishing their recombination with holes while facilitating the photoelectron transfer from the In₂O₃ CB to the Nb₂O₅ VB, thus resulting in no notable alteration in the lifetime (Fig. 4i). The analyses emphasize the ultrafast electron transfer at the In₂O₃/Nb₂O₅ S-scheme heterojunction interface for suppressing self-carrier recombination and spatially separating photoelectrons in the Nb₂O₅ CB and photoholes in the In₂O₃ VB.

Time-resolved fluorescence spectroscopy (TRPL) was conducted to investigate the long lifetime of the photocatalysts. As presented in Supplementary Fig. 15, under an Ar atmosphere, the IN10 hybrid exhibits a longer average lifetime ($\tau_a$) with respect to pristine In₂O₃ and Nb₂O₅ at an emission wavelength of 470 nm, where the fluorescence signals originate from both In₂O₃ and Nb₂O₅. Following the proposed S-scheme mechanism for IN10, photoelectrons in the In₂O₃ CB migrate

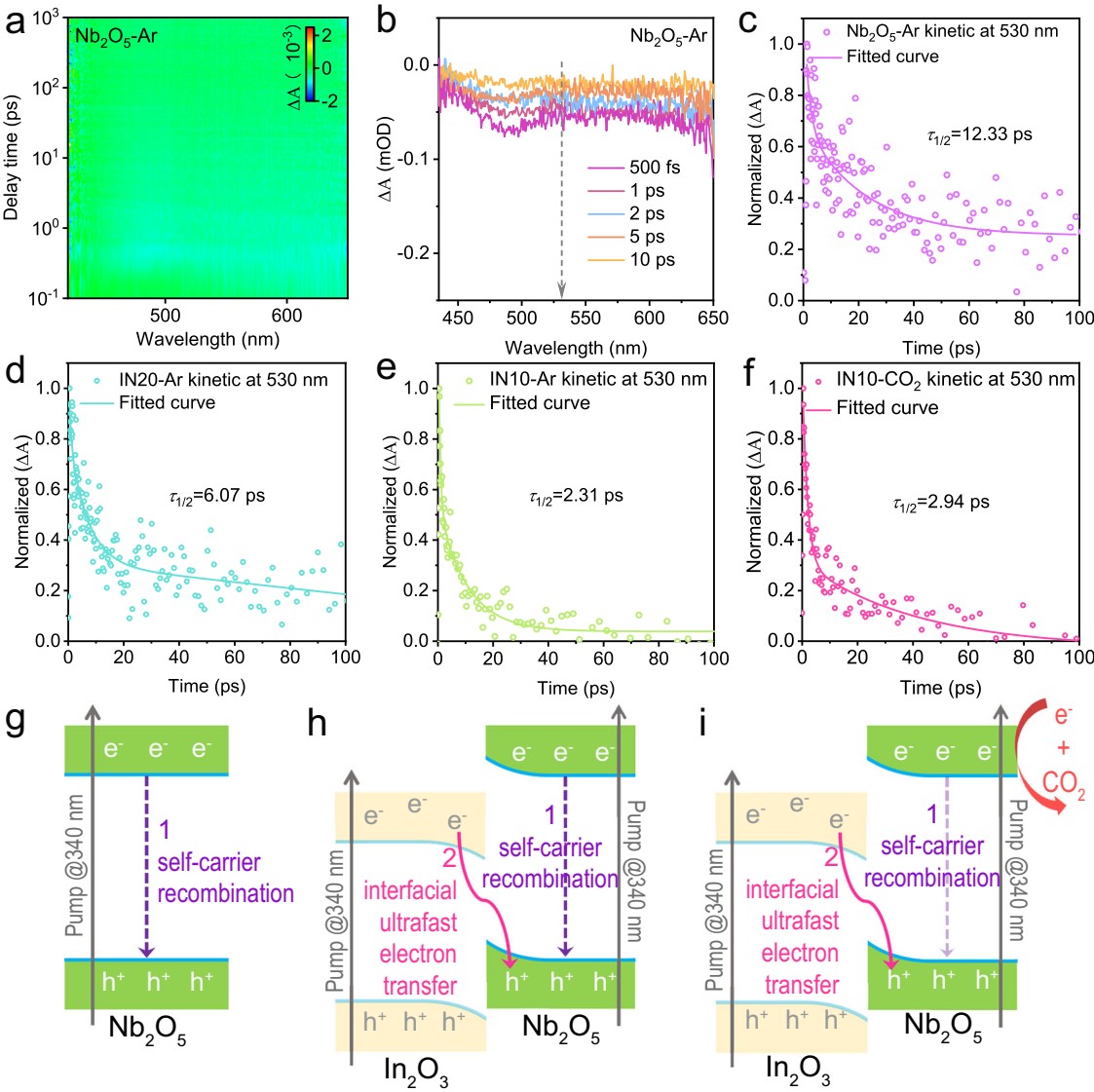

**Fig. 4 | Insights into charge transfer dynamics in pure $Nb_2O_5$ and $In_2O_3$/$Nb_2O_5$ S-scheme heterojunctions. a** The pseudocolor plot, and (**b**) transient absorption spectra of pure $Nb_2O_5$ recorded at indicated delay times measured with 340 nm excitation. Corresponding kinetic decay curves at 530 nm within 100 ps of (**c**) pure $Nb_2O_5$ in Ar, (**d**) IN20 in Ar, (**e**) IN10 in Ar, and (**f**) IN10 in $CO_2$. The decay pathways of photogenerated holes in (**g**) pure $Nb_2O_5$, (**h**) $In_2O_3$/$Nb_2O_5$ heterojunctions in Ar, and (**i**) $In_2O_3$/$Nb_2O_5$ heterojunctions in $CO_2$.

to the $Nb_2O_5$ VB and recombine with its photoholes, resulting in the accumulation of powerful electrons in the $Nb_2O_5$ CB and holes in the $In_2O_3$ VB, thereby prolonging the charge carrier lifetimes. The physically-mixed composite of $In_2O_3$ and $Nb_2O_5$ displays a shorter $\tau_a$ than the $In_2O_3$/$Nb_2O_5$ heterojunction, highlighting the significance of ultrafast interfacial electron transfer in extending carrier lifetimes. Furthermore, in situ TRPL was employed to explore the relationship between the ultrafast charge transfer-induced carrier lifetimes and the photocatalytic performance. The $\tau_a$ of IN10 recorded under a $CO_2$ atmosphere is shorter than that under an Ar atmosphere (Supplementary Fig. 16a), suggesting that a substantial portion of photogenerated electrons in the $Nb_2O_5$ CB is involved in $CO_2$ photoreduction, thereby leaving fewer charge carriers available for recombination. Bare $In_2O_3$, $Nb_2O_5$, and their physically-mixed composite (Supplementary Fig. 16b–d) reveal almost identical decay curves under both $CO_2$ and Ar atmospheres, indicating their poor photoreaction performance. Overall, the ultrafast interfacial charge transfer within the $In_2O_3$/$Nb_2O_5$ heterojunctions plays triple roles:

preventing the recombination of self-carriers, separating powerful photoelectrons and photoholes, and extending their long lifetimes.

## Chemisorption, activation and photoreduction of $CO_2$ over $In_2O_3$/$Nb_2O_5$ hybrid nanofibers

The adsorption and activation of $CO_2$ molecules on the catalyst, pivotal steps for $CO_2$ photoreaction, were investigated using DFT simulations and $CO_2$-temperature programmed desorption (TPD) analysis. Upon $CO_2$ adsorption on $Nb_2O_5$, distinct chemisorption processes occur, as evident from several observations (Fig. 5a–c and Supplementary Fig. 17): (i) a pronounced bending of the O=C=O bond at an angle of 127.4°; (ii) elongation of the bond length compared to free $CO_2$ molecule (1.16 Å); (iii) formation of new bonds between $CO_2$ and $Nb_2O_5$; and (iv) transfer of electrons from $Nb_2O_5$ to $CO_2$ (Supplementary Table 5). The integrated crystal orbital Hamiltonian population (ICOHP) of the C-O pairs is −18.37 and −13.82 eV in free and adsorbed $CO_2$, respectively, signifying $CO_2$ activation over $Nb_2O_5$. Moreover, the formation of new C-O (lattice) bonds with an ICOHP of

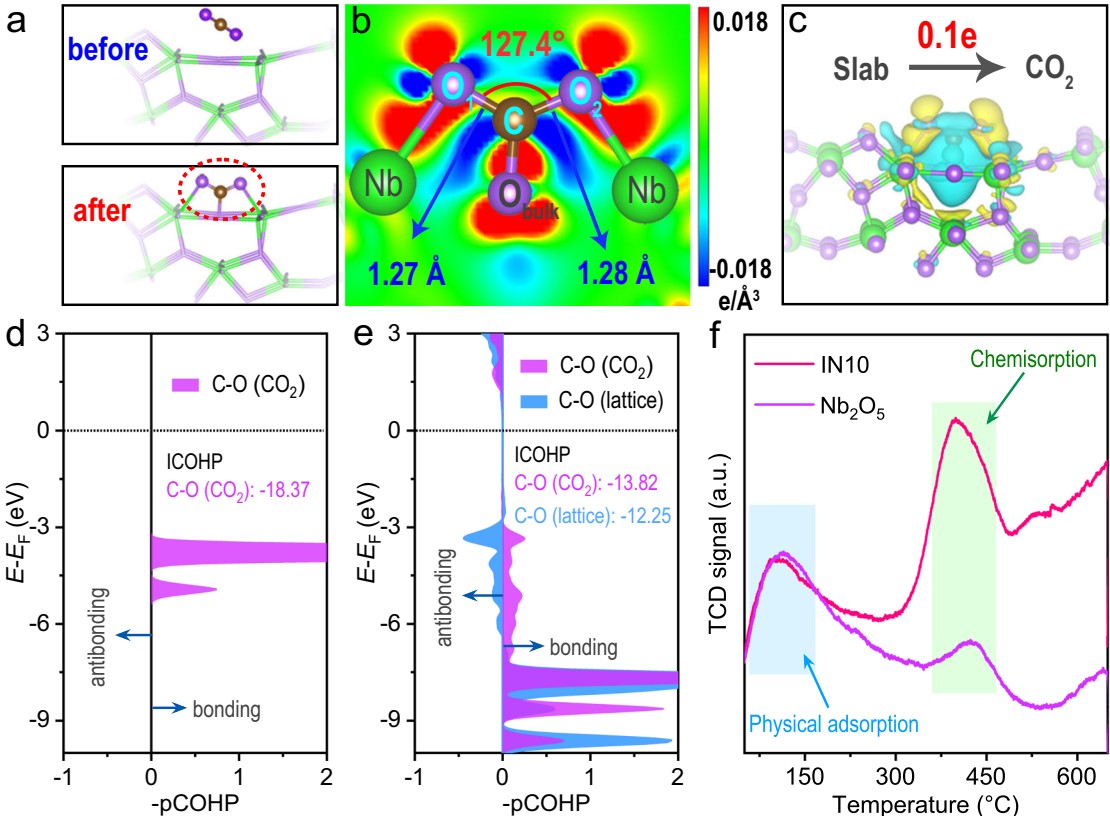

**Fig. 5 | Chemisorption and activation of CO₂ over In₂O₃/Nb₂O₅ hybrid nanofibers.** a, b Optimized structures, and (c) the corresponding charge density difference image of CO₂ adsorbed on the Nb₂O₅; cyan and yellow regions represent electron depletion and accumulation, respectively. The isosurface level is set to 0.002 e Å⁻³. Projected COHP profiles of (d) free CO₂ and (e) adsorbed CO₂ molecules. f CO₂-TPD spectra of pure Nb₂O₅ and IN10 composite.

−12.25 eV provides additional compelling evidence of the robust $CO_2$ chemisorption on $Nb_2O_5$ (Fig. 5d, e). The $CO_2$-TPD profiles for $Nb_2O_5$ and IN10 (Fig. 5f) demonstrate the desorption of physiosorbed $CO_2$ at 70–150 °C. At a high temperature ranging from 370 to 450 °C, both samples exhibit prominent desorption signals indicative of $CO_2$ chemisorption on the catalysts. The $CO_2$ adsorption energy ($E_{ads}$) on the Nb atom is more negative than that on the O atom (Supplementary Fig. 18 and Table 6), suggesting that Nb atoms serve as active sites for the chemisorption and activation of $CO_2$ during photoreduction.

The photocatalytic activities for $CO_2$ reduction of $In_2O_3$, IN$x$, and $Nb_2O_5$ were assessed in an online closed gas-circulation system (OLPCRS-2, Shanghai Boyi Scientific Instrument Co., Ltd., Supplementary Fig. 19) equipped with a glass reaction cell. Blank control experiments affirm that the concurrent presence of photocatalysts, $CO_2$, $H_2O$, and light irradiation is essential for initiating the photoreaction (Supplementary Fig. 20). In the absence of any molecule cocatalyst or scavenger, all the samples yielded CO as the reduction product with nearly 100% selectivity (Fig. 6a). Pure $In_2O_3$ and $Nb_2O_5$ exhibit poor photocatalytic performance due to the rapid recombination of photogenerated carriers inherent in single photocatalysts. However, the integration of $In_2O_3$ with $Nb_2O_5$ enhances $CO_2$ photoreduction activities, leading to a maximum CO production yield of 0.21 mmol g$_{active\ sites}^{-1}$ h$^{-1}$ over the IN10 composite. A comparison of $CO_2$ photoreduction performance was conducted among the $In_2O_3$/$Nb_2O_5$ hybrid nanofibers, the $In_2O_3$/$Nb_2O_5$ nanohybrid synthesized via the traditional dip-calcination method, and a physically-mixed composite of $In_2O_3$ and $Nb_2O_5$. This comparison emphasizes the critical importance of intimate interface contact between the two phases for facilitating ultrafast interfacial electron transfer within the

S-scheme heterojunction (Supplementary Fig. 21). Upon introducing tris(2,2′-bipyridyl)ruthenium(II) chloride hexahydrate ([Ru$^{II}$(bpy)₃] Cl₂·6H₂O) and 1,3-dimethyl-2-phenyl-2,3-dihydro-1H-benzo[d]imidazole (BIH) as the molecular catalyst and hole scavenger, respectively, a substantial amount of CO and a minor quantity of $H_2$ were detected, with the highest production yields (109.6 mmol g$_{active\ sites}^{-1}$ h$^{-1}$ for CO and 3.5 mmol g$_{active\ sites}^{-1}$ h$^{-1}$ for $H_2$) observed over the $In_2O_3$/$Nb_2O_5$ nanohybrid (Fig. 6b). To elucidate the origin of the photoreduction product, isotope-labeled carbon dioxide ($^{13}CO_2$) was employed as the substitute source gas for photocatalytic $CO_2$ reduction over IN10. Distinct peaks observed at 1.61 and 2.30 min in the total ion chromatography are assigned to $O_2$/Ar and $N_2$, respectively (Supplementary Fig. 22). Another prominent peak emerges at ~6.55 min, corresponding to CO, which generates the predominant mass spectrometry signal at $m/z$ = 29 ($^{13}CO$), accompanied by two additional fragments at $m/z$ = 13 and 16 ($^{13}C$ and O) (Fig. 6c). Additionally, thermogravimetric (TGA) analyses of pure $In_2O_3$, pristine $Nb_2O_5$, and the $In_2O_3$/$Nb_2O_5$ nanohybrid (IN10) reveal no perceptible weight changes up to 800 °C (Supplementary Fig. 23), suggesting that no carbon residual remains in the samples after a 2-h calcination at 600 °C. These findings confirm that the reduction product originates solely from the input $CO_2$, ruling out other potential carbon sources[59]. The recyclability and stability of IN10 for $CO_2$ photoreduction are confirmed, demonstrating a negligible decline in production yields over four cycles (Supplementary Fig. S24). The XRD pattern (Supplementary Fig. 25) and the In 3$d$ and Nb 3$d$ XPS spectra (Supplementary Fig. 26) of IN10 after the photoreaction exhibit inconspicuous changes compared to the fresh one, suggesting the photostability of the $In_2O_3$/$Nb_2O_5$ heterojunctions.

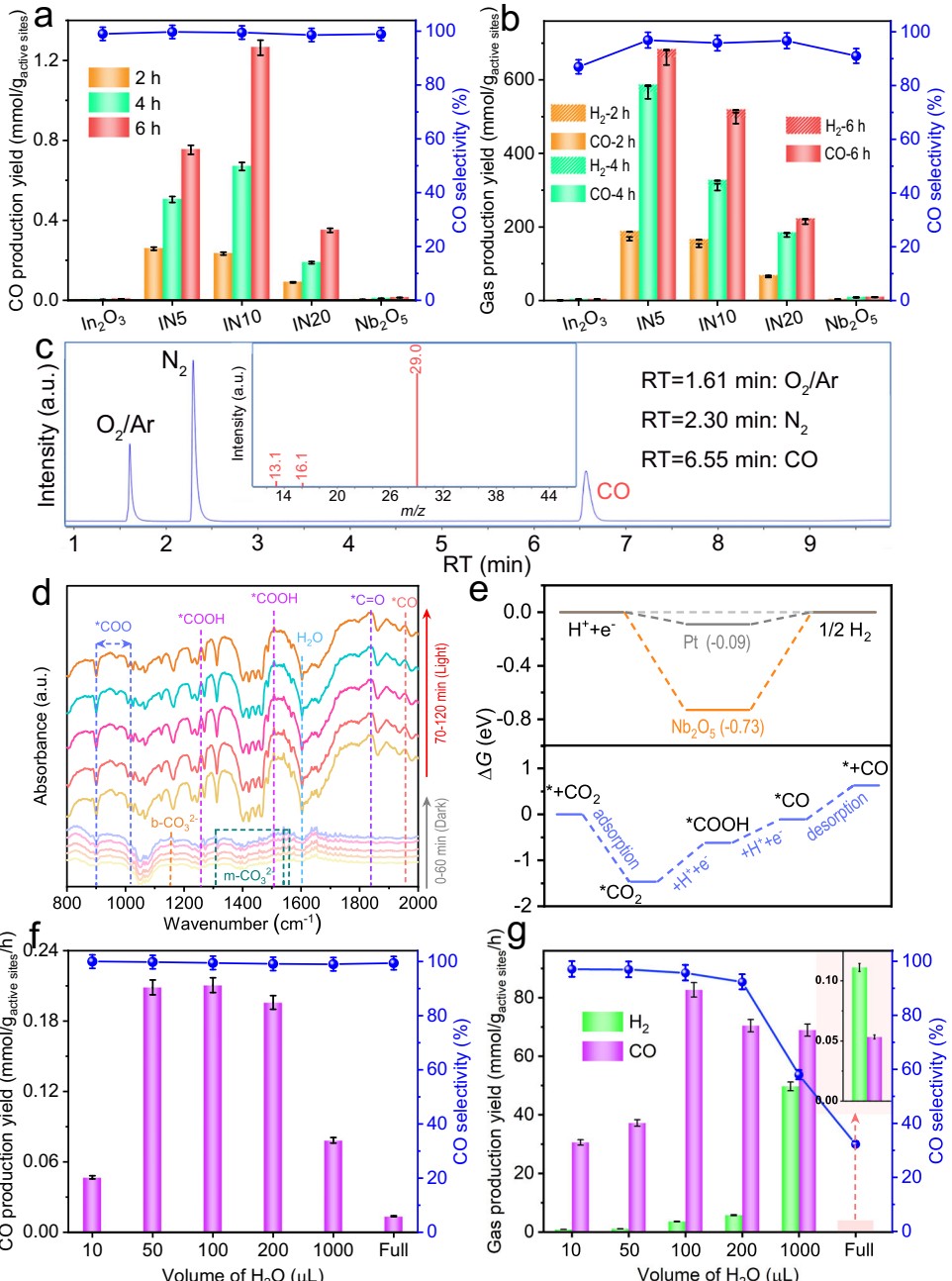

**Fig. 6 | Performance and mechanism insights into photocatalytic CO₂ reduction.** The production yields and CO selectivity over In₂O₃, INx, and Nb₂O₅ during six-hour experiments conducted under UV-visible light irradiation: **a** without any molecule cocatalyst or scavenger, (**b**) with [Ruᴵᴵ(bpy)₃]Cl₂·6H₂O and BIH. **c** The total ion chromatography and the corresponding mass spectra of the products in the photocatalytic reduction of ¹³CO₂ over IN10. **d** In situ DRIFT spectra for the photocatalytic CO₂ reduction over IN10. **e** Gibbs free energy diagrams of CO₂ photoreduction and H₂ production over Nb₂O₅ (100) slab. The influence of H₂O volumes on product selectivity over IN10: **f** without any molecule cocatalyst or scavenger, **g** with [Ruᴵᴵ(bpy)₃]Cl₂·6H₂O and BIH. The error bars (mean ± standard deviation) were obtained based on three independent photocatalytic experiments.

The reaction mechanism of CO₂ photoreduction was explored using both in situ diffuse reflectance infrared Fourier transform spectroscopy (DRIFTS, Supplementary Fig. 27) and DFT calculations. As depicted in Fig. 6d, the presence of bidentate carbonate (b-CO₃²⁻) and monodentate carbonate (m-CO₃²⁻) after the introduction of CO₂ into the system in the dark manifests the chemisorption of CO₂ on IN10. Under light irradiation, new adsorption bands of *COOH (1258 and 1507 cm⁻¹), *COO (carboxyl, 900 and 1017 cm⁻¹), *C=O (carbonyl, 1839 cm⁻¹), and *CO (absorbed CO, 1956 cm⁻¹) are detected, which are key intermediates in the conversion of CO₂ to CO. In light of these observations, the pathway for CO₂ photoreduction over In₂O₃/Nb₂O₅ hybrid nanofibers is proposed as follows, where * denotes the reaction active site[23,60,61]:

$$* + CO_2 \rightarrow {}^*CO_2 \tag{1}$$

$$^*CO_2 + H^+ + e^- \rightarrow {}^*COOH \tag{2}$$

$$^*COOH + H^+ + e^- \rightarrow {}^*CO + H_2O \tag{3}$$

$$^*CO \rightarrow {}^* + CO \tag{4}$$

DFT calculations reveal that the rate-limiting step for $CO_2$ reduction over $Nb_2O_5$ is the formation of *COOH intermediate. The adsorption of $CO_2$ molecules displays spontaneity with a decrease in Gibbs free energy, reaffirming the inevitable chemisorption and activation of $CO_2$ on $Nb_2O_5$. Based on the aforementioned analyses, the enhanced photocatalytic performance in $CO_2$ reduction achieved by the $In_2O_3/Nb_2O_5$ S-scheme heterojunctions can be attributed to several pivotal factors: (i) the close interconnection between $In_2O_3$ and $Nb_2O_5$; (ii) the S-scheme-induced ultrafast photoelectron transfer at the heterojunction interfaces; (iii) the effective separation of powerful photoelectrons in the $Nb_2O_5$ CB and photoholes in the $In_2O_3$ VB; (iv) the prolonged lifetimes of charge carriers within the nanohybrids; and (v) the $CO_2$ chemisorption and activation on $Nb_2O_5$.

In the liquid/solid photoreaction system, $H_2$ production from $H_2O$ reduction competes with CO generation from $CO_2$ reduction. To explore the influence of $H_2O$ content on product selectivity, different volumes of $H_2O$ were introduced into the reaction system. The results reveal an initial increase and subsequent decrease in CO production yield with the gradual addition of $H_2O$ (Fig. 6f). At a low $H_2O$ volume, fewer protons ($H^+$) are available for $CO_2$ reduction, resulting in poor performance. Conversely, excessive $H_2O$ in the reaction solvent diminishes the solubility of $CO_2$ and impedes its activation on the catalyst. Notably, significant $H_2$ production is absent in the absence of molecular catalysts and hole scavengers, regardless of $H_2O$ volume. Indeed, the high $H_2$-evolution barrier over $Nb_2O_5$ poses a challenge for $H_2$ production (Fig. 6e)[62]. Even if a trace amount of $H_2$ is generated, it can be consumed by the residual $O_2$ within the system, originating from input high-purity $CO_2$ (99.999%), following an exothermic reaction ($H_2 + 1/2O_2 \rightarrow H_2O$, $\Delta G < 0$). On the other hand, the CO selectivity is affected by the $H_2O$ content in the reaction system involving $[Ru^{II}(bpy)_3]Cl_2 \cdot 6H_2O$ and BIH, exhibiting a noticeable decrease when its volume exceeds 200 μL (Fig. 6g). This suggests that the input $CO_2/H_2O$ can precisely tune the composition of the photoreduction products.

In a $CO_2$ photoreduction system without hole sacrificial agents, two simultaneous processes govern the overall amount of $O_2$ within the system: $O_2$ generation from $H_2O$ photooxidation ($H_2O + 2h^+ \rightarrow 1/2O_2 + 2H^+$, OER) and its consumption through photoreduction ($O_2 + 4e^- + 4H^+ \rightarrow 2H_2O$, ORR). Over time, all the samples manifest a decline in $O_2$ levels (Supplementary Fig. 28), indicating a higher rate of $O_2$ consumption compared to its production. Free energy diagrams revel that the OER is not a spontaneous reaction for both $In_2O_3$ and $Nb_2O_5$, while the ORR is a spontaneous reaction (Supplementary Figs. 29 and 30), suggesting a preference for $O_2$ consumption over its generation and, consequently, a substantial reduction in $O_2$ quantity in the reaction system.

In summary, S-scheme $In_2O_3/Nb_2O_5$ hybrid nanofibers were synthesized using a facile one-step electrospinning method, establishing intimate phase contact for seamless charge carrier transport. Fs-TAS analyses confirmed the ultrafast electron transfer at the $In_2O_3/Nb_2O_5$ S-scheme heterojunction interface, involving the recombination of feeble photoelectrons in the $In_2O_3$ CB and holes in the $Nb_2O_5$ VB, while efficiently separating and preserving powerful photoelectrons in the $Nb_2O_5$ CB and holes in the $In_2O_3$ VB for participation in photoreactions. Both DFT calculations and $CO_2$-TPD results demonstrated the efficient chemisorption and activation of $CO_2$ molecules on the $In_2O_3/Nb_2O_5$ heterojunctions. Benefiting from the prolonged lifetimes driven by the rapid interfacial charge transfer and efficient $CO_2$ activation on the catalyst, the optimized $In_2O_3/Nb_2O_5$ nanofibers exhibited enhanced $CO_2$-reduction performance, yielding CO with an impressive output of up to 0.21 mmol $g_{active\ sites}^{-1}$ $h^{-1}$ in the absence of any molecule cocatalyst or scavenger. This work highlights the potential of advanced fs-TAS techniques in exploring ultrafast charge transfer at S-scheme heterojunction interfaces.

## Methods

### Chemicals
All the chemicals are of analytical grade (AR) and were used without further purification. Indium nitrate hydrate ($In(NO_3)_3 \cdot xH_2O$, 99.9%) and ammonium niobium oxalate (V) hydrate ($C_4H_4NNbO_9 \cdot nH_2O$, 99.9%) were purchased from Macklin Biochemical Technology Co., Ltd. (Shanghai, China). Polyvinylpyrrolidone (PVP, $M_W = 1300,000$) and tris(2,2′-bipyridyl)ruthenium(II) chloride hexahydrate ($[Ru^{II}(bpy)_3]Cl_2 \cdot 6H_2O$, 98%) were purchased from Aladdin Biochemical Technology Co., Ltd. (Shanghai, China). N,N-dimethylformamide (DMF, AR) and acetonitrile (AR) were obtained from Sinopharm Chemical Reagent Co., Ltd. (Shanghai, China). 1,3-dimethyl-2-phenyl-2,3-dihydro-1H-benzo[d] imidazole (BIH) was synthesized according to our previous work[34].

### Synthesis of $In_2O_3/Nb_2O_5$ (IN$x$) hybrid nanofibers
The $In_2O_3/Nb_2O_5$ hybrid nanofibers were synthesized via a one-step electrospinning method by mixing the precursors of both phases in the same electrospinning solution. Typically, 0.32 g of $In(NO_3)_3 \cdot xH_2O$ and 1.50 g of PVP were dissolved in 10 mL of DMF and stirred at room temperature until a clear solution was obtained. Simultaneously, 0.032 g of $C_4H_4NNbO_9 \cdot nH_2O$, corresponding to 10 wt.% of $Nb_2O_5$ relative to $In_2O_3$ (IN10), was dissolved in 1 mL of $H_2O$. This solution was then added to the previously prepared one and stirred for 2 h. Subsequently, the viscous solution was loaded into a syringe equipped with a stainless-steel nozzle, positioned about 10 cm away from the collector. An electric potential of 20 kV was applied, and the solution was fed with a rate of 0.4 mL $h^{-1}$. The collected nanofibrous mat was calcined at 600 °C for 2 h with a heating rate of 2 °C $min^{-1}$ to completely remove the PVP, resulting in the formation of the $In_2O_3/Nb_2O_5$ nanofibers with a yield of over 90%. For comparison, $In_2O_3/Nb_2O_5$ heterojunctions with different $Nb_2O_5$ contents were synthesized by changing the amount of $C_4H_4NNbO_9 \cdot nH_2O$ to 0.016 and 0.065 g, resulting in nominal weight percentages of 5 wt.% and 20 wt.% $Nb_2O_5$ relative to $In_2O_3$ (IN5 and IN20), respectively.

### Synthesis of pure $In_2O_3$ nanofibers
Pure $In_2O_3$ nanofibers were synthesized by dissolving 0.16 g of $In(NO_3)_3 \cdot xH_2O$ in 5 mL of DMF. Once the indium nitrate was completely dissolved, 0.75 g of PVP was added, and the mixture was stirred until the solution became clear. This solution was then loaded into a syringe equipped with a stainless-steel nozzle, positioned approximately 10 cm from the collector. A potential of 20 kV was applied, and the solution was fed at a rate of 0.4 mL $h^{-1}$. The samples were collected and calcined at 600 °C for 2 h with a heating rate of 2 °C $min^{-1}$ to obtain pure $In_2O_3$ nanofibers with a yield of approximately 90%.

### Synthesis of pure $Nb_2O_5$ nanorods
Typically, 0.08 g of $C_4H_4NNbO_9 \cdot nH_2O$ was first dissolved in 1 mL of $H_2O$. Once fully dissolved, 5 mL of DMF and 0.75 g of PVP were added while stirring until the solution was clear. This solution was then loaded into a syringe equipped with a stainless-steel nozzle, positioned about 10 cm from the collector. An electric potential of 20 kV was applied, and the solution was fed at a rate of 0.6 mL $h^{-1}$. After calcining at 600 °C for 2 h with a heating rate of 2 °C $min^{-1}$, pure $Nb_2O_5$ nanorods were obtained with a yield of over 90%.

### Photocatalytic $CO_2$ reduction
The $CO_2$ photoreduction was carried out in an online gas-closed system equipped with a gas-circulated pump (OLPCRS-2, Shanghai Boyi Scientific Instrument Co., Ltd.). Typically, 10 mg of photocatalysts, 30 mL of acetonitrile, and 100 μL of $H_2O$ were added into a glass reactor connected to the online system. The airtight system underwent complete evacuation using a vacuum pump. Then, ~60 kPa of high-purity $CO_2$ (99.999%) gas was injected. After adsorption equilibrium, a 300 W Xe arc lamp (Microsolar 300 Xenon lamp source, Beijing

Perfectlight, China) was used as the light source without any filter. The reaction system was maintained at 8 °C, controlled by cooling water. The gas chromatograph (GC-2030, Shimadzu Corp., Japan) equipped with barrier discharge ionization detector (BID) and a capillary column (Carboxen 1010 PLOT Capillary, 60 m × 0.53 mm) was employed to analyze the photocatalytic $CO_2$ reduction products. For comparison, 2 mM of $[Ru^{II}(bpy)_3]Cl_2·6H_2O$ and 10 mM of BIH were introduced into the photoreaction system, with other parameters unchanged. Regarding active sites, pure $In_2O_3$ employs itself as its active sites, while the IN$x$ and pure $Nb_2O_5$ utilize the $Nb_2O_5$ as active sites. The value of $m_{active\ sites}$ in all $In_2O_3/Nb_2O_5$ nanohybrids was determined based on the actual weight ratios of $Nb_2O_5$ within the composites (Supplementary Table 1). Given that the CB minimum and the VB maximum of the $In_2O_3/Nb_2O_5$ heterojunction are predominantly contributed by Nb 4$d$ and $O_{In2O3}$ 2$p$ orbitals (Supplementary Fig. 31), it follows that $CO_2$ photoreduction and $H_2O$ photooxidation occur specifically over the Nb and $O_{In2O3}$ atoms, respectively[63].

The isotope-labeling experiment was conducted using $^{13}CO_2$ (isotope purity, 99%, and chemical purity, 99.9%) as the carbon source. The gas products were analyzed by gas chromatography-mass spectrometry (8890 GC System, 5977B GC/MSD, Agilent Technologies, USA) equipped with the column for detecting the reduction products (HP-MOLESIEVE). Helium was used as carrier gas. The temperatures of the injector and EI source were set to be 150 and 200 °C, respectively.

### Statistics and reproducibility
No statistical method was used to predetermine sample size. No data were excluded from the analyses. The experiments were not randomized, and we were not blinded to allocation during experiments and outcome assessment.

## Data availability
The source data underlying Figs. 1f, g, 2a–d, 3d–j, 4b–f, 5d–f, 6, and Supplementary Figs. 3–12, 13b-c, 14–17, 20, 21, 23–26, 28–31a are provided as a Source Data file, which is available in figshare with the identifier https://doi.org/10.6084/m9.figshare.25844110 and in the Source Data file. All data are available from the corresponding author on request. Source data are provided with this paper.

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

## Acknowledgements

This work was supported by the National Key Research and Development Program of China (2022YFE0115900 (J.Y.) and 2022YFB3803600 (J.Y.)), National Natural Science Foundation of China (22378371 (F.X.), 22361142704 (J.Y.), 52003213 (F.X.), 22238009 (J.Y.), 51932007 (J.Y.) and 22261142666 (J.Y.)), China Postdoctoral Science Foundation (2022M712958 (F.X.)), the Natural Science Foundation of Hubei Province of China (2022CFA001 (F.X.)), and the Fundamental Research Funds for the Central Universities, China University of Geosciences (Wuhan) (No.CUG22061 (J.Y.)). Partial support of Iran National Science Foundation (Grant Number 4021464 (J.Y.)) was acknowledged. We also thanked the Faculty of Materials Science and Chemistry, China University of Geosciences (CUG), Wuhan for its TEM facilities and the data analysis of Dr. Mingxing Gong.

## Author contributions

F.X. and J.Y. conceived and designed the experiments. X.D. carried out the synthesis of the materials, the photocatalytic test, and the characterizations of the materials. J.Z. and G.L. performed the ultrafast TA measurements. X.D., J.Z., and K.Q. analyzed all the results. X.D. wrote the manuscript. F.X. conducted the DFT calculations, contributed to data analysis, and revised the manuscript. F.X. and J.Y. supervised the project. All authors discussed the results and commented on the manuscript.

## Competing interests

The authors declare no competing interests.

## Additional information

**Supplementary information** The online version contains
supplementary material available at

Feiyan Xu or Jiaguo Yu.

**Peer review information** *Nature Communications* thanks the anon-
ymous reviewers for their contribution to the peer review of this work. A
peer review file is available.

