## [Peer Review File · Nature Communications]

Ultrafast Electron Transfer at the In₂O₃/Nb₂O₅ S-scheme Interface for CO₂ PhotoreductionREVIEWER COMMENTS

Reviewer #1 (Remarks to the Author):

The authors report an S-scheme heterojunction by integrating Nb₂O₅ with In₂O₃ nanofibers for CO₂ photoreduction and conducted a series of experiments to examine the S-scheme charge separation mechanism by using multiple techniques, especially femtosecond transient absorption spectroscopy (fs-TAS). The charge dynamics analyses suggest the presence of an ultrafast photoelectron transfer from In₂O₃ conduction band to Nb₂O₅ valence band, separating robust photoelectrons/holes and prolonging their lifetimes. This finding is meaningful as it provides direct experimental evidence for the ultrafast charge separation dynamics in such S-scheme heterojunctions for artificial photosynthesis. However, certain claims in the current version still need further explanation or additional experimental support. I recommend publishing the paper after the authors appropriately address the following issues.

1. The main finding involves the observation of ultrafast electron transfer at the interfaces of the In₂O₃/Nb₂O₅ S-scheme heterojunction. This implies the significance of interfacial contact between the two phases. To bolster this finding, I recommend the authors conduct a comparative fs-TAS analysis on physically-mixed composites of In₂O₃ and Nb₂O₅ to explore potential differences, which would be insightful to underscore the advantages of S-scheme heterojunctions.
2. In the HRTEM image presented in Figure 1c, both (211) and (222) planes are observed in the In₂O₃ nanofibers. However, all the DFT calculations focus on the In₂O₃ (111) surface. It would be beneficial for readers to gain a clearer understanding of the criteria that influenced the choice of In₂O₃ (111) facets for DFT calculations.
3. I would recommend that the authors provide additional details on the methodology used to determine the band structures of In₂O₃ and Nb₂O₅. This could include a more elaborate explanation of the experimental techniques or computational methods employed in the analysis.
4. To enhance the understanding of in situ DRIFTS in connection with the experimental reaction system, it is recommended to include detailed information on experimental parameters such as gas pressure, flow rate, temperature, and the instrumentation used.

Reviewer #2 (Remarks to the Author):

In this manuscript, novel In₂O₃/Nb₂O₅ S-scheme heterojunctions were fabricated through a straightforward one-step electrospinning technique. The In₂O₃/Nb₂O₅ hybrid nanofibers demonstrated remarkable improvement in photocatalytic CO₂ conversion. The enhanced performance was attributed to ultrafast interfacial electron transfer, powerful redox ability, efficient chemisorption and activation of CO₂ molecules. The results and findings are intriguing and impactful. I recommend the publication of the

paper with minor revisions.

1. In Fig. 5c, the charge density difference image should include a color code indicating electron gain or loss. There appears to be apparent electron gains for CO₂, according to the authors' claim, which should be derived from Bader charge analyses. If so, please provide the atomic Bader charge of each atom of the adsorbed CO₂. Besides, what is the isosurface value in this image?
2. Fig. 2 shows the geometric structure and work function of In₂O₃ and Nb₂O₅. For work function calculations, it is crucial to construct a surface slab with a sufficient number of atomic layers. However, the atomic layers shown in Fig. 2c, d appear somewhat limited. To enhance accuracy, please check work function convergence by varying atomic layers for In₂O₃ and Nb₂O₅.
3. The study observed the production yields of a majority of CO and a minority of H₂, indicating a relatively high selectivity for CO. Is the minor quantity of H₂ product correlated with the low content of water vapor in the liquid/solid reaction? Can the composition of the product be tuned by adjusting the ratio of input CO₂/H₂O? Exploring this aspect could make this study more interesting.
4. Regarding the evaluation of photocatalytic performance using a 300 W Xenon lamp (Microsolar 300 Xenon Lamp Source, Beijing Perfect Light, China), clarification is needed on whether a filter was used.
5. In the transient absorption test, the author selected excitation light with a wavelength of 340 nm to excite Nb₂O₅ and In₂O₃. Please provide a brief explanation for choosing this specific wavelength.
6. The experimental protocol for the in situ DRIFT measurement is currently absent and should be included.

Reviewer #3 (Remarks to the Author):

In this manuscript, the authors report the selective CO₂ photo-reduction catalyzed by In₂O₃/Nb₂O₅ hybrid nanofibers, with an impressive output to CO (18.6 mmol g⁻¹). The catalyst is well characterized, providing basis for in-depth study of the reaction mechanism. Using a facile one-step electrospinning method achieves the catalysis synthesis. There are also many results that need to be further discussed. Especially the stability and the structure of the photocatalyst in working conditions. Besides, the mechanism of how the photocatalyst works is also doubtful, which seriously influences the novelty of this work. A few concerns are detailed below.

1. Regarding the innovativeness of the manuscript, according to the prevalent opinion, the S-scheme heterojunctions established intimate contact between the two phases for seamless charge carrier transport and enhanced chemisorption, activation and photoreduction of CO₂ (doi.org/10.1016/j.chempr.2020.06.010). The mechanism of the S-scheme heterojunctions catalyst having a facilitating effect on the reaction is a known process. The catalyst prepared in the article then

mechanistically repeats the proof of the already existing points. What is the difference between this work and the literature?

2. Regarding the determination of CO₂ reaction sites, how can the different reaction sites be distinguished when both In₂O₃ and Nb₂O₅ catalyse the hydrogenation of CO₂ to CO with activity? How to correlate the reaction sites?

3. Does CO₂ adsorption change during light and are the calculations reasonable?

4. By controlling the proportion of precursors added, it is recommended to add ICP etc., to quantify whether the catalysts are indeed prepared at that proportion.

5. The innovation of the article is that the faster the electron transfer after the formation of the S-scheme heterojunctions is more favourable for the reaction, which can be illustrated by modulating the number of heterojunctions by catalyst preparation, and further correlations can be formed through the lifetime of transient absorption.

6. Regarding the article reaction design, using 1,3-dimethyl-2-phenyl-2,3-dihydro-1H-benzo[d]imidazole (BIH) as a hole scavenger, is it possible to couple an oxidation reaction to improve the utilisation of light energy? How can the hydrogen-containing gas mixture obtained be well utilised?

7. Regarding product analysis, are liquid phase products detected?

8. Regarding the determination of the bandgap structure of catalysts, the determination of Fermi energy levels with flat band potentials is not accurate, and it is suggested to add UPS to determine the positions of the energy levels more accurately; the positions of the Fermi energy levels in the bandgap structure have been marked in Supplementary Figure 4d, and regarding how to obtain the bandgap CB and VB, please specify the formulae and the calculation process.

Reviewer #4 (Remarks to the Author):

Deng et al reported S-scheme In₂O₃/Nb₂O₅ photocatalysts for CO₂ reduction and utilized the fs-transient absorption spectra, etc to investigate the interfacial charge dynamics of the S-scheme heterojunction. However, there still lack solid evidences to draw the conclusions on the charge dynamics and catalytic mechanism. Moreover, the photocatalytic conversion rate along with CO₂ reduction selectivity of as-designed photocatalytic system tightly depends on the photosensitizer and so on, which weakens its superiority compared with reported systems. Based above, unless the authors solve all the concerns, or the manuscript is not suitable to be published anywhere.

1.To achieve an intimate interface, the In₂O₃/Nb₂O₅ photocatalysts were synthesized by the one-step electrospinning method. Relevant synthetic process mechanism related to the final morphology should be illustrated. To better highlight the advantage of this method, a reference sample via traditional method might be provided. In addition, it is noticed PVP is one of the raw materials, and it has been reported to capable of anticipating the catalytic processes. How to exclude its influences?

2.The authors are suggested to demonstrate the superiority of the photocatalytic system. The advantage of S-scheme systems is that the the thermodynamic energies could be designed to be sufficient for the oxidation and reduction reactions. Therefore, why is the system still dependent on the photosentizer and

sacrificial agent? Or the authors just consider to demonstrate a high photocatalytic performance? However, the addition of these additives might confuse the charge separation mechanism.

3. Following the last comment, the conditions for collecting the TAS spectra is totally different with the reaction conditions. Therefore, the conclusion drawn from the current TAS spectra might be correlated with the photocatalytic performances without adding any additive. While it is found that without additives, the main product is H₂ instead of CO. Based above, the correlation between the comparison of the TAS spectra in Ar and CO₂ and the real photocatalytic reaction with additives is not logical and reasonable at all.

4. For the 2D TAS data in Figure 3(a, b, and c), the effect of chirping has not been calibrated. It is recommended that the authors provide calibrated data to enable a more accurate assessment of the dynamics.

5. In the parameterization of the kinetic fitting, the proportional relationship between different time components is equally important for the overall lifetime analysis. Therefore, it is recommended that the authors provide the corresponding proportions of A₁, A₂, A₃, etc., and re-evaluate the ultrafast kinetic decay lifetime in a scientifically rigorous manner.

6. In the spectral discussion of the S-type heterojunction, the authors mentioned the accumulation of electrons in the Nb₂O₅ conduction band. Therefore, please provide corresponding TAS spectroscopic evidence and the analysis to support this claim.

7. The fs-TAS testing can reflect the ultrafast dynamics at the heterojunction interface. However, the subsequent catalytic reaction occurs on a much slower timescale, ranging from ms to s. There is a temporal mismatch between these two processes. Therefore, based on the conclusions drawn from Figure 4, it doesn't make sense about the ultrafast reaction kinetics involved in CO₂ reduction. Additionally, please assess whether there is a direct relationship between ultrafast dynamics and the photocatalytic performance.

8. As for the verification of the catalytic mechanism, DFT calculation and in-situ FTIR were adopted. Still, if Nb₂O₅ could benefit the catalytic CO₂ reduction, why H₂ is the main product when no additives introduced. This proves that tiny water might be preferentially reduced compared with CO₂. In addition, the change of in-situ FTIR spectra is not obvious. But according to the high photocatalytic performances indicated in this work, the FTIR spectra change should be quite obvious.

9. English This approach ensures clarity and coherence in the article while employing precise and concise language to convey ideas.

Response to Reviewers' comments

Reviewer #1:

The authors report an S-scheme heterojunction by integrating Nb₂O₅ with In₂O₃ nanofibers for CO₂ photoreduction and conducted a series of experiments to examine the S-scheme charge separation mechanism by using multiple techniques, especially femtosecond transient absorption spectroscopy (fs-TAS). The charge dynamics analyses suggest the presence of an ultrafast photoelectron transfer from In₂O₃ conduction band to Nb₂O₅ valance band, separating robust photoelectrons/holes and prolonging their lifetimes. This finding is meaningful as it provides direct experimental evidence for the ultrafast charge separation dynamics in such S-scheme heterojunctions for artificial photosynthesis. However, certain claims in the current version still need further explanation or additional experimental support. I recommend publishing the paper after the authors appropriately address the following issues.

1. The main finding involves the observation of ultrafast electron transfer at the interfaces of the In₂O₃/Nb₂O₅ S-scheme heterojunction. This implies the significance of interfacial contact between the two phases. To bolster this finding, I recommend the authors conduct a comparative fs-TAS analysis on physically-mixed composites of In₂O₃ and Nb₂O₅ to explore potential differences, which would be insightful to underscore the advantages of S-scheme heterojunctions.

Response: The fs-TAS analysis of physically-mixed composite of In₂O₃ and Nb₂O₅ reveals longer lifetimes (τ_1 and τ_2) compared to In₂O₃/Nb₂O₅ heterostructures (Supplementary Fig. 12), indicating inefficient electron transfer from In₂O₃ to Nb₂O₅ and underscoring the advantages of interfacial contact between the two phases in the S-scheme heterojunction for efficient charge transfer and separation.

The related results and discussion have been included in the revised manuscript.

2. In the HRTEM image presented in Figure 1c, both (211) and (222) planes are observed in the In₂O₃ nanofibers. However, all the DFT calculations focus on the In₂O₃ (111) surface. It would be beneficial for readers to gain a clearer understanding of the criteria that influenced the choice of In₂O₃ (111) facets for DFT calculations.

Response: The In₂O₃ nanofibers consist of quasi-spherical tiny grains without anisotropic shapes, which tend to expose surfaces with lower surface energy driven by the energy minimization during high-temperature calcination process. As depicted in Supplementary Table 2, the (111) surface of In₂O₃ demonstrates the lowest surface energy among the typical low-index facets, which is consistent with previous report (*ACS Catal.* 2021, 11, 1406-1423). Consequently, we opted to use the In₂O₃ (111) surface for our DFT calculations.

Supplementary Table 2. The calculated surface energy of various facets of In₂O₃.

Facet	Surface energy (J m ⁻²)
-------	-------------------------------------

100	2.02
110	1.06
111	0.74
201	1.65

The surface energy γ is defined as the energy per unit area required to form the surface relative to the bulk. It is calculated using the following formula:

$$(S1)$$

Here, U_{slab} represents the total energy of the relaxed surface slab with a 20 Å vacuum region, U_{bulk} denotes the energy of the equivalent bulk In_2O_3 units, and A represents the surface area created on each side of the surface slab. During the calculation of surface energy, the middle layer of the surface slab was fixed, while the top and bottom layers were allowed to relax.

3. I would recommend that the authors provide additional details on the methodology used to determine the band structures of In_2O_3 and Nb_2O_5 . This could include a more elaborate explanation of the experimental techniques or computational methods employed in the analysis.

Response: According to the ultraviolet photoelectron spectroscopy (UPS) spectra (Supplementary Fig. 6a-b), the valance band (VB) maximum of In_2O_3 and Nb_2O_5 is estimated at 2.41 and 2.28 V (*vs.* SHE), respectively. Combining with the bandgap values disclosed in Supplementary Fig. 6c, the conduct band (CB) minimum is established as -0.49 V for In_2O_3 and -0.96 V for Nb_2O_5 (Supplementary Fig. 6d).

The UPS experimental protocols and the band structure calculation have been included in the revised manuscript.

4. To enhance the understanding of in situ DRIFTS in connection with the experimental reaction system, it is recommended to include detailed information on experimental parameters such as gas pressure, flow rate, temperature, and the instrumentation used.

Response: In situ diffuse reflectance infrared Fourier transform spectra (DRIFTS) were acquired on the Nicolet iS50 spectrometer (Thermo Scientific, USA) equipped with a specialized reactor (Supplementary Fig. 23). Prior to measurement, samples were compressed into cylinder shapes with a diameter of 0.6 cm under 10 MPa. The experimental procedure involved two sequential stages in a continuous-flow mode. Initially, CO_2 was purged into the chamber with saturated water vapor at a flow rate of 20 mL min^{-1} for 60 minutes in the absence of light to explore the CO_2 adsorption on the photocatalyst. Subsequently, a 365-nm LED light was activated for another 60 minutes to investigate the photoreaction intermediates.

Reviewer #2:

In this manuscript, novel $\text{In}_2\text{O}_3/\text{Nb}_2\text{O}_5$ S-scheme heterojunctions were fabricated through a

straightforward one-step electrospinning technique. The $\text{In}_2\text{O}_3/\text{Nb}_2\text{O}_5$ hybrid nanofibers demonstrated remarkable improvement in photocatalytic CO_2 conversion. The enhanced performance was attributed to ultrafast interfacial electron transfer, powerful redox ability, efficient chemisorption and activation of CO_2 molecules. The results and findings are intriguing and impactful. I recommend the publication of the paper with minor revisions.

1. In Fig. 5c, the charge density difference image should include a color code indicating electron gain or loss. There appears to be apparent electron gains for CO_2 , according to the authors' claim, which should be derived from Bader charge analyses. If so, please provide the atomic Bader charge of each atom of the adsorbed CO_2 . Besides, what is the isosurface value in this image?

Response: The cyan and yellow regions in Fig. 5c represent electron depletion and accumulation, respectively. The isosurface level is set to $0.002 \text{ e } \text{\AA}^{-3}$. The atomic Bader charge of the adsorbed CO_2 on the Nb_2O_5 surface was provided in Supplementary Table 4. The result shows the adsorbed CO_2 gain 0.1 electrons from Nb_2O_5 , consistent with the analysis of the charge density difference.

The related information has been included in the revised manuscript.

2. Fig. 2 shows the geometric structure and work function of In_2O_3 and Nb_2O_5 . For work function calculations, it is crucial to construct a surface slab with a sufficient number of atomic layers. However, the atomic layers shown in Fig. 2c, d appear somewhat limited. To enhance accuracy, please check work function convergence by varying atomic layers for In_2O_3 and Nb_2O_5 .

Response: We assessed the convergence of the work function by varying the number of atomic layers for In_2O_3 and Nb_2O_5 . As illustrated in Fig. 2c, d and Supplementary Fig. 4-5, the work function of In_2O_3 (111) and Nb_2O_5 (100) converges to 5.71 and 5.65 eV, respectively. Notably, irrespective of the number of layers applied, the work function (Φ) of In_2O_3 (111) consistently remained higher than that of Nb_2O_5 (100), which forces the electron transfer from Nb_2O_5 to In_2O_3 upon their hybridization in the S-scheme heterojunctions.

3. The study observed the production yields of a majority of CO and a minority of H_2 , indicating a relatively high selectivity for CO. Is the minor quantity of H_2 product correlated with the low content of water vapor in the liquid/solid reaction? Can the composition of the product be tuned by adjusting the ratio of input $\text{CO}_2/\text{H}_2\text{O}$? Exploring this aspect could make this study more interesting.

Response: In the liquid/solid photoreaction system, H_2 production from H_2O reduction competes with CO generation from CO_2 reduction. To explore the influence of H_2O content on product selectivity, varying volumes of H_2O were introduced into the reaction system, revealing an initial increase and subsequent decrease in CO production yield with the gradual addition of H_2O (Fig. 6f). At a low H_2O volume, fewer protons (H^+) are available for CO_2 reduction, resulting in poor performance. Conversely, excessive H_2O in the reaction solvent diminishes the solubility of CO_2 and impedes its activation on the catalyst, thereby reducing overall performance. Notably, no significant H_2 is observed in the absence of molecular catalysts and hole scavengers, regardless of changes in H_2O volume. Indeed, the high H_2 -evolution barrier over Nb_2O_5 poses a challenge for

H₂ production (Fig. 6e). Even if a trace amount of H₂ is generated, it can be consumed by the residual O₂ within the system originating from input high-purity CO₂ (99.999%), which follows an exothermic reaction ($\text{H}_2 + 1/2\text{O}_2 \rightarrow \text{H}_2\text{O}$, $\Delta G < 0$). On the other hand, the H₂O content significantly influences CO selectivity as the reaction system involves [Ru^{II}(bpy)₃]Cl₂·6H₂O and BIH, exhibiting a noticeable decrease when its volume exceeds 200 μL (Fig. 6g). This suggests that the input CO₂/H₂O can precisely tune the composition of the photoreduction products.

The related results and discussion have been incorporated in the revised manuscript.

4. Regarding the evaluation of photocatalytic performance using a 300 W Xenon lamp (Microsolar 300 Xenon Lamp Source, Beijing Perfect Light, China), clarification is needed on whether a filter was used.

Response: A 300 W Xe arc lamp (Microsolar 300 Xenon lamp source, Beijing Perfectlight, China) was used as the light source without any filter.

5. In the transient absorption test, the author selected excitation light with a wavelength of 340 nm to excite Nb₂O₅ and In₂O₃. Please provide a brief explanation for choosing this specific wavelength.

Response: Based on the UV-vis spectra (Fig. 1f), employing excitation light with a wavelength of 340 nm proves effective in photoexciting both In₂O₃ and Nb₂O₅, resulting in the generation of photoinduced electrons and holes.

6. The experimental protocol for the in situ DRIFT measurement is currently absent and should be included.

Response: In situ diffuse reflectance infrared Fourier transform spectra (DRIFTS) were acquired on the Nicolet iS50 spectrometer (Thermo Scientific, USA) equipped with a specialized reactor (Supplementary Fig. 23). Prior to measurement, samples were compressed into cylinder shapes with a diameter of 0.6 cm under 10 MPa. The experimental procedure involved two sequential stages in a continuous-flow mode. Initially, CO₂ was purged into the chamber with saturated water vapor at a flow rate of 20 mL min⁻¹ for 60 minutes in the absence of light to explore the CO₂ adsorption on the photocatalyst. Subsequently, a 365-nm LED light was activated for another 60 minutes to investigate the photoreaction intermediates.

Reviewer #3:

In this manuscript, the authors report the selective CO₂ photo-reduction catalyzed by In₂O₃/Nb₂O₅ hybrid nanofibers, with an impressive output to CO (18.6 mmol g⁻¹). The catalyst is well characterized, providing basis for in-depth study of the reaction mechanism. Using a facile one-step electrospinning method achieves the catalysis synthesis. There are also many results that need to be further discussed. Especially the stability and the structure of the photocatalyst in working conditions. Besides, the mechanism of how the photocatalyst works is also doubtful,

which seriously influences the novelty of this work. A few concerns are detailed below.

1. Regarding the innovativeness of the manuscript, according to the prevalent opinion, the S-scheme heterojunctions established intimate contact between the two phases for seamless charge carrier transport and enhanced chemisorption, activation and photoreduction of CO₂ (doi.org/10.1016/j.chempr.2020.06.010). The mechanism of the S-scheme heterojunctions catalyst having a facilitating effect on the reaction is a known process. The catalyst prepared in the article then mechanistically repeats the proof of the already existing points. What is the difference between this work and the literature?

Response: The S-scheme heterojunctions, integrating both reduction and oxidation photocatalysts, have proven effective in spatially separating photogenerated charge carriers with robust redox capabilities. Conventionally, the construction of S-scheme heterojunctions involves initially acquiring photocatalyst I and subsequently applying photocatalyst II onto I through methods such as in situ growth or electrostatic self-assembly. However, this post-hybridization method cannot ensure the maximum contact area between two phases at atomic levels, thus impeding the efficient interfacial transport of photogenerated carriers and compromising photocatalytic efficiency (Fig. 1a).

Here we designed a desirable S-scheme heterojunction by mixing the precursors of both phases in the same electrospinning solution. As a result, In₂O₃ and Nb₂O₅ formed simultaneously during the high-temperature calcination of the electrospinning nanofibers. Such “one-pot” preparation method assures to maximize the phase contact at atomic levels, which provides an unimpeded transport route and promotes interfacial charge transfer between In₂O₃ and Nb₂O₅ (Fig. 1a). Owing to the intimate phase contact, we observed an ultrafast photoelectron transfer from In₂O₃ CB to Nb₂O₅ VB by using the advanced femtosecond transient absorption spectroscopy (fs-TAS), which help to inhibit self-carrier recombination, segregate powerful photoelectrons and prolong the long-lifetimes of the nanohybrids.

In summary, we prepared the In₂O₃/Nb₂O₅ S-scheme heterojunctions using a "one-pot-like" method, which maximizes atomic phase contact among the heterojunctions and accelerates ultrafast photoelectron transfer, as directly observed by fs-TAS. Our findings could advance the preparation of more efficient S-scheme heterojunctions and deepen our understanding of the ultrafast S-scheme photoelectron transfer process.

2. Regarding the determination of CO₂ reaction sites, how can the different reaction sites be distinguished when both In₂O₃ and Nb₂O₅ catalyse the hydrogenation of CO₂ to CO with activity? How to correlate the reaction sites?

Response: Owing to the S-scheme charge separation (Fig. 2e), the photoelectrons accumulate on the Nb₂O₅ CB while holes accumulate on the In₂O₃ VB. Accordingly, the Nb₂O₅ in the heterojunctions is the active site for CO₂ reduction. As for bare In₂O₃, itself behaves as its active sites. The value of $m_{\text{active sites}}$ in all In₂O₃/Nb₂O₅ nanohybrids is determined based on the actual weight ratios of Nb₂O₅ within the composites (Supplementary Table 1).

3. Does CO₂ adsorption change during light and are the calculations reasonable?

Response: The CO₂ chemisorption on the catalyst in the dark were confirmed by both DFT simulations and CO₂-TPD analysis, which are reasonable and credible. Under light irradiation, these activated CO₂ molecules engage in photoreactions to produce reduction products. However, neither DFT calculations nor CO₂-TPD experiments can be performed under light irradiation, preventing us from gaining insights into the changes of CO₂ adsorption during light exposure.

4. By controlling the proportion of precursors added, it is recommended to add ICP etc., to quantify whether the catalysts are indeed prepared at that proportion.

Response: The precise Nb₂O₅ content in all composites was determined using inductively coupled plasma-atomic emission spectrometry (ICP-AES), and the results are presented in Supplementary Table 1.

5. The innovation of the article is that the faster the electron transfer after the formation of the S-scheme heterojunctions is more favourable for the reaction, which can be illustrated by modulating the number of heterojunctions by catalyst preparation, and further correlations can be formed through the lifetime of transient absorption.

Response: Thanks for your positive feedback. Regarding the correlation between electron transfer rates and catalyst composition, our findings reveal intriguing insights.

On the one hand, in an Ar atmosphere, both τ_1 and τ_2 lifetimes demonstrate a gradual decrease with increasing Nb₂O₅ content (IN5 and IN10, Fig. 3h, i). This implies the rapid migration of more photoelectrons from the In₂O₃ CB to Nb₂O₅ upon hybridization, resulting in a smaller number of electrons available for diffusion and trapping processes (Fig. 3k, l).

On the other hand, under an Ar atmosphere, the half-life ($\tau_{1/2}$) of both IN20 and IN10 is shorter than that of pure Nb₂O₅, signifying the migration of photoelectrons from the In₂O₃ CB to Nb₂O₅, thereby reducing the number of photoholes in the VB (Fig. 4h). Notably, In₂O₃/Nb₂O₅ heterojunctions exhibit a composition-dependent half-life, with shorter values as Nb₂O₅ content decreases (IN10 < IN20). According to the S-scheme charge separation mechanism, photoelectrons in the In₂O₃ CB transfer to the Nb₂O₅ VB and recombine with its photoholes in equal proportions. At an ideal In₂O₃/Nb₂O₅ ratio (*e.g.*, IN10), the population of photogenerated charge carriers in both materials is approximately identical, leaving few excess photoholes in the Nb₂O₅ VB and, consequently, a shorter lifetime after S-scheme charge separation. When Nb₂O₅ content deviates from its ideal composition (*i.e.*, IN20), some excessive photoholes remain in the Nb₂O₅ VB, leading to a longer lifetime than IN10.

These results and discussion have been included in the revised manuscript.

6. Regarding the article reaction design, using 1,3-dimethyl-2-phenyl-2,3-dihydro-1H-benzo[d]imidazole (BIH) as a hole scavenger, is it possible to couple an oxidation reaction to improve the utilisation of light energy? How can the hydrogen-containing gas mixture obtained be well utilised?

Response: We reassessed the photocatalytic activities for CO₂ reduction of all the samples both

in the absence of any molecule cocatalyst or scavenger and upon introducing $[\text{Ru}^{\text{II}}(\text{bpy})_3]\text{Cl}_2 \cdot 6\text{H}_2\text{O}$ and BIH into the system.

In the absence of any molecule cocatalyst or scavenger, the reduction product of all the samples was identified as CO with nearly 100% selectivity (Fig. 6a). Pure In_2O_3 and Nb_2O_5 exhibit poor photocatalytic performance due to the rapid recombination of photogenerated carriers inherent in single photocatalysts. However, the integration of In_2O_3 with Nb_2O_5 enhances CO_2 photoreduction activities, leading to a noteworthy maximum CO production yield of $0.21 \text{ mmol g}_{\text{active sites}}^{-1} \text{ h}^{-1}$ over the IN10 composite. A comparison of CO_2 photoreduction performance among the $\text{In}_2\text{O}_3/\text{Nb}_2\text{O}_5$ hybrid nanofibers, $\text{In}_2\text{O}_3/\text{Nb}_2\text{O}_5$ nanohybrid synthesized *via* the traditional method, as well as the physically-mixed composite of In_2O_3 and Nb_2O_5 emphasizes the critical role of the S-scheme heterojunction with intimate contact between the two phases for ultrafast interfacial electron transfer (Supplementary Fig. 17). Upon introducing $[\text{Ru}^{\text{II}}(\text{bpy})_3]\text{Cl}_2 \cdot 6\text{H}_2\text{O}$ and BIH as the molecular catalyst and hole scavenger, respectively, a substantial amount of CO and a minor quantity of H_2 were detected, with the highest production yields ($109.6 \text{ mmol g}_{\text{active sites}}^{-1} \text{ h}^{-1}$ for CO and $3.5 \text{ mmol g}_{\text{active sites}}^{-1} \text{ h}^{-1}$ for H_2) observed over the $\text{In}_2\text{O}_3/\text{Nb}_2\text{O}_5$ nanohybrid (Fig. 6b).

In the liquid/solid photoreaction system, H_2 production from H_2O reduction competes with CO generation from CO_2 reduction. To explore the influence of H_2O content on product selectivity, varying volumes of H_2O were introduced into the reaction system, revealing an initial increase and subsequent decrease in CO production yield with the gradual addition of H_2O (Fig. 6f). At a low H_2O volume, fewer protons (H^+) are available for CO_2 reduction, resulting in poor performance. Conversely, excessive H_2O in the reaction solvent diminishes the solubility of CO_2 and impedes its activation on the catalyst, thereby reducing overall performance. Notably, no significant H_2 is observed in the absence of molecular catalysts and hole scavengers, regardless of changes in H_2O volume. Indeed, the high H_2 -evolution barrier over Nb_2O_5 poses a challenge for H_2 production (Fig. 6e)⁵⁸. Even if a trace amount of H_2 is generated, it can be consumed by the residual O_2 within the system, originating from input high-purity CO_2 (99.999%), which follows an exothermic reaction ($\text{H}_2 + 1/2\text{O}_2 \rightarrow \text{H}_2\text{O}$, $\Delta G < 0$). On the other hand, H_2O content significantly influences CO selectivity as the reaction system involves $[\text{Ru}^{\text{II}}(\text{bpy})_3]\text{Cl}_2 \cdot 6\text{H}_2\text{O}$ and BIH, exhibiting a noticeable decrease when its volume exceeds $200 \mu\text{L}$ (Fig. 6g). This suggests that the input $\text{CO}_2/\text{H}_2\text{O}$ can precisely tune the composition of the photoreduction products.

In utilizing hydrogen-containing mixtures, we see two potential applications. Firstly, pure H_2 can be separated from H_2/CO mixtures using the Pressure Swing Adsorption (PSA) method, widely adopted in industry. Secondly, appropriately balanced H_2/CO mixtures can be directly utilized as synthesis gas (syngas) in hydrocarbon production.

These results and discussion have been updated in the revised manuscript. In our future research, we aim to integrate CO_2 photoreduction with an oxidation reaction improve the overall utilization of light energy.

7. Regarding product analysis, are liquid phase products detected?

Response: The liquid phase was analyzed using both a liquid chromatograph (LC-2030 Plus,

Shimadzu Corp., Japan) and a gas chromatograph (GC-2014C, Shimadzu Corp., Japan) equipped with an HP-Plot Q column (Agilent, America), revealing no detectable products or intermediates were found.

8. Regarding the determination of the bandgap structure of catalysts, the determination of Fermi energy levels with flat band potentials is not accurate, and it is suggested to add UPS to determine the positions of the energy levels more accurately; the positions of the Fermi energy levels in the bandgap structure have been marked in Supplementary Figure 4d, and regarding how to obtain the bandgap CB and VB, please specify the formulae and the calculation process.

Response: According to the ultraviolet photoelectron spectroscopy (UPS) spectra (Supplementary Fig. 6a-b), the VB maximum of In_2O_3 and Nb_2O_5 is estimated at 2.41 and 2.28 V (vs. SHE), respectively. Combining with the bandgap values disclosed in Supplementary Fig. 6c, the CB minimum is established as -0.49 V for In_2O_3 and -0.96 V for Nb_2O_5 (Supplementary Fig. 6d).

The calculation of the band structures based on UPS spectra for In_2O_3 and Nb_2O_5 has been included in the revised manuscript.

Reviewer #4:

Deng et al reported S-scheme $\text{In}_2\text{O}_3/\text{Nb}_2\text{O}_5$ photocatalysts for CO_2 reduction and utilized the fs-transient absorption spectra, etc to investigate the interfacial charge dynamics of the S-scheme heterojunction. However, there still lack solid evidences to draw the conclusions on the charge dynamics and catalytic mechanism. Moreover, the photocatalytic conversion rate along with CO_2 reduction selectivity of as-designed photocatalytic system tightly depends on the photosensitizer and so on, which weakens its superiority compared with reported systems. Based above, unless the authors solve all the concerns, or the manuscript is not suitable to be published anywhere.

1. To achieve an intimate interface, the $\text{In}_2\text{O}_3/\text{Nb}_2\text{O}_5$ photocatalysts were synthesized by the one-step electrospinning method. Relevant synthetic process mechanism related to the final morphology should be illustrated. To better highlight the advantage of this method, a reference sample via traditional method might be provided. In addition, it is noticed PVP is one of the raw materials, and it has been reported to capable of anticipating the catalytic processes. How to exclude its influences?

Response: (1) Electrospinning stands out as a widely employed technique for the production of one-dimensional nanofibers through electrostatic forces. The high-voltage power supply connects to both the conducting collector and the needle, which contains a liquid solution comprising $\text{In}(\text{NO}_3)_3 \cdot x\text{H}_2\text{O}$, $\text{C}_4\text{H}_4\text{NNbO}_9 \cdot n\text{H}_2\text{O}$, and PVP. This connection creates an electric field between the needle tip and the collector, propelling the charged droplet suspended on the needle tip towards the collector once the applied high voltage surpasses the droplet's surface tension. This triggers the formation of a Taylor cone, and the charged solution transforms into a jet that travels

toward the oppositely charged collector. During the trajectory of the jet, solvent evaporation occurs, and charged fibers accumulate on the conducting collector. As a result, the charges on the fibers dissipate, leading to the formation of a nonwoven nanofibrous mat. Subsequently, the obtained nanofibrous mat undergoes calcination at 600 °C for 2 h with a heating rate of 2 °C min⁻¹ to completely remove PVP, resulting in the formation of the In₂O₃/Nb₂O₅ nanofibers.

(2) We have made a performance comparison among the In₂O₃/Nb₂O₅ nanofibers, In₂O₃/Nb₂O₅ nano hybrids synthesized traditionally, and the physically mixed composite of In₂O₃ and Nb₂O₅. This comparison emphasizes the critical role of intimate contact between the two phases for the ultrafast interfacial electron transfer within S-scheme heterojunctions (Supplementary Fig. 17).

(3) The electrospun nanofibrous precursors underwent high-temperature calcination in air to form the resulting oxide heterojunctions nanofibers. During this process, both PVP and residual solvent were completely removed. To confirm this, we conducted thermogravimetric (TGA) analyses of In₂O₃, Nb₂O₅, and In₂O₃/Nb₂O₅ nano hybrid (IN10). The TGA results showed no perceptible weight changes up to a temperature of 800 °C (Supplementary Fig. 19), suggesting that no carbon residual remains in the samples after a 2-hour calcination at 600 °C.

These results and discussion have been incorporated in the revised manuscript.

2. The authors are suggested to demonstrate the superiority of the photocatalytic system. The advantage of S-scheme systems is that the thermodynamic energies could be designed to be sufficient for the oxidation and reduction reactions. Therefore, why is the system still dependent on the photosensitizer and sacrificial agent? Or the authors just consider to demonstrate a high photocatalytic performance? However, the addition of these additives might confuse the charge separation mechanism.

Response: Thanks for the reviewer's professional comments. The initial purpose of incorporating a photosensitizer and sacrificial agent was to accentuate the distinct properties of various samples, emphasizing the influence of S-scheme mechanism, interfacial ultrafast charge transfer, and other factors on photocatalytic activity. Taking into account the concerns raised by the reviewer, we have systematically reevaluated the performance without any photosensitizer and sacrifice agent. The corresponding results have been included in the revised manuscript.

3. Following the last comment, the conditions for collecting the TAS spectra is totally different with the reaction conditions. Therefore, the conclusion drawn from the current TAS spectra might be correlated with the photocatalytic performances without adding any additive. While it is found that without additives, the main product is H₂ instead of CO. Based above, the correlation between the comparison of the TAS spectra in Ar and CO₂ and the real photocatalytic reaction with additives is not logical and reasonable at all.

Response: (1) Fs-TAS was employed to investigate the dynamic process of photogenerated carriers, spanning from excitation generation to transfer and separation. The introduction of a CO₂ atmosphere aims to explore the dynamic changes of charge carriers in the actual photocatalysis process, serving to validate the proposed S-scheme photocatalytic mechanism. However, it is emphasized that this simulation does not replicate the authentic CO₂

photoreduction process. The reduction of CO₂ using photogenerated electrons is a time-consuming process that cannot be completed at the picosecond scale.

(2) We checked the raw data and identified a slight fluctuation in the baseline around the signal of H₂ in the chromatogram, which initially led to a mistaken interpretation of H₂ production. To further confirm the photocatalytic performance, we re-synthesized fresh catalysts, performed performance tests with greater precision, and systematically studied the effect of the amount of H₂O added to the catalytic system on the CO and H₂ yields both in the absence of molecular cocatalysts and scavengers and in their presence. The related results and discussion have been included in the revised manuscript.

4. For the 2D TAS data in Figure 3(a, b, and c), the effect of chirping has not been calibrated. It is recommended that the authors provide calibrated data to enable a more accurate assessment of the dynamics.

Response: We have thoroughly rechecked all the TAS data in the manuscript. All the transient data have been chirp-corrected, ensuring the accuracy of the dynamics assessment.

5. In the parameterization of the kinetic fitting, the proportional relationship between different time components is equally important for the overall lifetime analysis. Therefore, it is recommended that the authors provide the corresponding proportions of A₁, A₂, A₃, etc., and re-evaluate the ultrafast kinetic decay lifetime in a scientifically rigorous manner.

Response: Thank you for your valuable suggestion. We have meticulously re-evaluated the ultrafast kinetic decay lifetime of all relevant samples, providing the lifetimes (τ_1, τ_2, τ_3), as well as the corresponding proportions of A₁, A₂ and A₃ in the revised manuscript (Fig. 3g-j).

6. In the spectral discussion of the S-type heterojunction, the authors mentioned the accumulation of electrons in the Nb₂O₅ conduction band. Therefore, please provide corresponding TAS spectroscopic evidence and the analysis to support this claim.

Response: The broad GSB signal of Nb₂O₅ appears around 500 nm (Fig. 4a, b), offering valuable insights into the population of photoholes in its VB, as affirmed by the disappearance of the signal following the introduction of a hole-trapping agent (lactic acid) (Supplementary Fig. 13). To minimize the interference of In₂O₃, kinetic decay curves of bare Nb₂O₅ and the In₂O₃/Nb₂O₅ nano hybrids (IN20 and IN10) are derived at 530 nm (Fig. 4c-e), mainly involving two processes related to photoexcited holes in the Nb₂O₅ VB: recombination with self-generated photoelectrons (process 1) and recombination with photoelectrons transferred from In₂O₃ (process 2) (Fig. 4g). Under an Ar atmosphere, the half-life ($\tau_{1/2}$) of both IN20 and IN10 is shorter than that of pure Nb₂O₅, signifying the migration of photoelectrons from the In₂O₃ CB to Nb₂O₅, thereby reducing the number of photoholes in the VB (Fig. 4h). Notably, In₂O₃/Nb₂O₅ heterojunctions exhibit a composition-dependent half-life, with shorter values as Nb₂O₅ content decreases (IN10 < IN20). According to the S-scheme charge separation mechanism, photoelectrons in the In₂O₃ CB transfer to the Nb₂O₅ VB and recombine with its photoholes in equal proportions. At an ideal In₂O₃/Nb₂O₅ ratio (*e.g.*, IN10), the population of photogenerated charge carriers in both materials

is approximately identical, leaving few excess photoholes in the Nb₂O₅ VB and, consequently, a shorter lifetime after S-scheme charge separation. When Nb₂O₅ content deviates from its ideal composition (*i.e.*, IN20), some excessive photoholes remain in the Nb₂O₅ VB, leading to a longer lifetime than IN10.

As the GSB signal of Nb₂O₅ corresponds to the photoholes in its VB, the accumulation of photoelectrons in the Nb₂O₅ CB within the In₂O₃/Nb₂O₅ S-scheme heterojunction can only be indirectly deduced based on the charge transfer dynamics. Nevertheless, we have supplemented this analysis with EPR spectra of DMPO-•O₂⁻ and •OH species to confirm the accumulation of photogenerated electrons and holes after S-scheme charge separation. The reduction potential of •O₂⁻ and the oxidation potential of •OH are -0.74 and 2.28 V (*vs.* SHE), respectively. As anticipated, in contrast to pristine In₂O₃ and Nb₂O₅, the signals of both •O₂⁻ and •OH significantly intensify in the In₂O₃/Nb₂O₅ composite (Supplementary Fig. 10). This observation signifies the efficient separation and accumulation of highly energetic photoelectrons in the Nb₂O₅ CB and photoholes in the In₂O₃ VB, providing compelling evidence for the S-scheme charge separation mechanism.

7. The fs-TAS testing can reflect the ultrafast dynamics at the heterojunction interface. However, the subsequent catalytic reaction occurs on a much slower timescale, ranging from ms to s. There is a temporal mismatch between these two processes. Therefore, based on the conclusions drawn from Figure 4, it doesn't make sense about the ultrafast reaction kinetics involved in CO₂ reduction. Additionally, please assess whether there is a direct relationship between ultrafast dynamics and the photocatalytic performance.

Response: The photocatalytic reaction process typically involves photoexcitation, carrier recombination or migration, and active participation in redox reactions. Fs-TAS was employed to investigate the dynamic process of photogenerated carriers, spanning from excitation generation to transfer and separation. The introduction of a CO₂ atmosphere aims to explore the dynamic changes of charge carriers in the actual photocatalysis process, serving to validate the proposed S-scheme photocatalytic mechanism. However, it is emphasized that this simulation does not replicate the authentic CO₂ photoreduction process. The reduction of CO₂ using photogenerated electrons is a time-consuming process that cannot be completed at the picosecond scale.

8. As for the verification of the catalytic mechanism, DFT calculation and in-situ FTIR were adopted. Still, if Nb₂O₅ could benefit the catalytic CO₂ reduction, why H₂ is the main product when no additives introduced. This proves that tiny water might be preferentially reduced compared with CO₂. In addition, the change of in-situ FTIR spectra is not obvious. But according to the high photocatalytic performances indicated in this work, the FTIR spectra change should be quite obvious.

Response: (1) We checked the raw data and identified a slight fluctuation in the baseline around the signal of H₂ in the chromatogram, which initially led to a mistaken interpretation of H₂ production. To further confirm the photocatalytic performance, we re-synthesized fresh catalysts, performed performance tests with greater precision, and systematically studied the effect of the

amount of H₂O added to the catalytic system on the CO and H₂ yields both in the absence of molecular cocatalysts and scavengers and in their presence.

(2) In response to the Reviewer's suggestion, we have further refined the in situ DRIFTS testing method. Typically, in situ DRIFTS were acquired on the Nicolet iS50 spectrometer (Thermo Scientific, USA) equipped with a specialized reactor (Supplementary Fig. 23). Prior to measurement, samples were compressed into cylinder shapes with a diameter of 0.6 cm under 10 MPa. The experimental procedure involved two sequential stages in a continuous-flow mode. Initially, CO₂ was purged into the chamber with saturated water vapor at a flow rate of 20 mL min⁻¹ for 60 minutes in the absence of light to explore the CO₂ adsorption on the photocatalyst. Subsequently, a 365-nm LED light was activated for another 60 minutes to investigate the photoreaction intermediates.

As depicted in Fig. 6d, the presence of bidentate carbonate (b-CO₃²⁻) and monodentate carbonate (m-CO₃²⁻) after the introduction of CO₂ into the system in the dark, further manifesting the chemisorption of CO₂ on the IN10. Under light irradiation, new adsorption bands of *COOH (1258 and 1507 cm⁻¹), *COO (carboxyl, 900 and 1017 cm⁻¹), *C=O (carbonyl, 1839 cm⁻¹) and *CO (adsorbed carbon monoxide, 1956 cm⁻¹) are detected, which are pivotal intermediates in the conversion of CO₂ to CO.

9. English This approach ensures clarity and coherence in the article while employing precise and concise language to convey ideas.

Response: In the revised manuscript, we have polished the English accordingly. Many thanks.

REVIEWER COMMENTS

Reviewer #1 (Remarks to the Author):

The authors answered my questions very well and I am very satisfied with their revisions and recommend publication.

Reviewer #2 (Remarks to the Author):

We thank the reviewers for the response. The authors answered all the queries properly. Hence it is accepted for publication.

Reviewer #3 (Remarks to the Author):

Please find the detailed comments in the attached.

Reviewer #4 (Remarks to the Author):

After checking the revised manuscript and the comments, I think the authors have made reasonable responses to the proposed comments and proper revision to the original one, thus it is suitable to be publishable in the currently-revised manuscript.

Response to Reviewer' comments

Reviewer #1:

The authors answered my questions very well and I am very satisfied with their revisions and recommend publication.

Response: Thanks for your positive feedback.

Reviewer #2:

We thank the reviewers for the response. The authors answered all the queries properly. Hence it is accepted for publication.

Response: Many thanks.

Reviewer #3:

Deng *et al.* reported the photocatalysis for the selective CO₂ photo-reduction catalyzed by In₂O₃/Nb₂O₅ hybrid nanofibers, with an impressive output to CO. They achieved an impressive performance for CO production (18.6 mmol g⁻¹). The authors ascribed the high activity for CO production to S-scheme heterojunction by using the “one-pot” preparation method, which ensures maximizing the phase contact at atomic levels. After revision, the content of the article is more complete. However, no evidence supports the presence of the reaction site, and no direct evidence was provided to support the idea that CO₂ adsorption is enhanced under light irradiation. Moreover, “one-pot” preparation method is not an innovative way to prepare catalysts. Therefore, this work does not meet the high standards of *Nature Communication*.

1. Evidences for reaction site are lacking.

In Response to Reviewers' comments, the author replied that “Owing to the S-scheme charge separation (Fig. 2e), the photoelectrons accumulate on the Nb₂O₅ CB while holes accumulate on the In₂O₃ VB. Accordingly, the Nb₂O₅ in the heterojunctions is the active site for CO₂ reduction. As for bare In₂O₃, itself behaves as its active sites”. It is not advisable to assign the reaction site through the band gap structure. For example, the reduction site is on Nb₂O₅. What Nb₂O₅ site is it specifically? The oxidation site is on In₂O₃, where is it specifically? Is it possible that the catalyst is located close to the interface? In addition, regarding the correlation between reaction and site, the author replied: “The value of m_{active} sites in all In₂O₃/Nb₂O₅ nanohybrids is determined based on the actual weight ratios of Nb₂O₅ within the composites (Supplementary Table 1).”, it is recommended to conduct a more detailed correlation between catalyst structure and reaction activity. Also, how do charge separation and electron transfer specifically relate to reactivity? Is it

possible to provide in-situ or quasi-situ transient absorption energy support? All in all, the reaction site of the catalyst was not well-characterized, not to mention if the acid and base sites are correlated with the catalytic activity of CO formation from CO₂ and methanol.

Response: (1) In the In₂O₃/Nb₂O₅ S-scheme heterostructure, photoelectrons in the In₂O₃ CB are transferred to the Nb₂O₅ VB and recombine with its photoholes, resulting in the accumulation of powerful photoelectrons and holes on the Nb₂O₅ CB and the In₂O₃ VB, respectively, for participation in photoreactions. This charge transfer route and mechanism are strongly supported by in situ irradiated XPS, fs-TAS, EPR, *et al.* In this case, when the In₂O₃/Nb₂O₅ heterojunctions are employed as photocatalysts, CO₂ molecules can only react with photoelectrons enriched in the Nb₂O₅ CB, indicating that Nb₂O₅ in the heterojunctions serves as the active site for CO₂ reduction. Given that the CB minimum and the VB maximum of the In₂O₃/Nb₂O₅ heterojunction are predominantly contributed by Nb 4*d* and O_{In₂O₃} 2*p* orbitals (Supplementary Fig. 29), it follows that CO₂ photoreduction and H₂O photooxidation occur specifically over the Nb and O_{In₂O₃} atoms, respectively.

On the other hand, upon CO₂ adsorption on the Nb₂O₅, distinct chemisorption processes occur, as evident from several observations (Fig. 5a-c and Supplementary Fig. 16): *i*) a pronounced bending of the O=C=O bond at an angle of 127.4°; *ii*) elongation of the bond length with respect to that of the free CO₂ molecule (1.16 Å); *iii*) formation of new bonds between CO₂ and Nb₂O₅; *iv*) transfer of electrons from Nb₂O₅ to CO₂ (Supplementary Table 4). Furthermore, the adsorption energy (E_{ads}) of CO₂ molecules on the Nb atom is more negative compared to that on the O atom (Supplementary Fig. 17 and Table 5). This observation underscores the role of Nb atoms as the specifically active sites, effectively chemisorbing and activating CO₂ molecules, thus facilitating the photocatalytic CO₂ reduction reactions.

These results and discussion have been updated in the revised manuscript.

(2) As illustrated in Fig. 1a, *the unique structure of the In₂O₃/Nb₂O₅ heterojunction endows close contact between the two phases without any hindrance, promoting ultrafast interfacial photoelectron transfer from In₂O₃ to Nb₂O₅ and prolonging carrier lifetimes, thereby enhancing the photocatalytic performance.* During the photocatalytic reaction, these processes are interconnected and progressively develop to achieve the desired results. Consequently, we believe that the catalyst's structure plays a crucial role in boosting photocatalytic activity.

(3) The photocatalytic reaction process typically involves three key stages: photoexcitation, carrier recombination or migration, and active participation in redox reactions. Although carrier transfer, separation, or recombination occurs within nanoseconds or even shorter timescales, and CO₂ photoreduction reactions take place over milliseconds or seconds, a vital connection exists among these processes. *Efficient separation of photogenerated carriers with strong redox ability is essential for prolonging carrier lifetimes to enhance photocatalytic activity, with ultrafast electron transfer at the heterojunction interfaces playing a crucial role in carrier separation.* Thus, we believe that *ultrafast interfacial electron transfer constitutes the fundamental mechanism underlying the enhancement of photocatalytic performance.* In response to the

reviewer's comments, we endeavored to explore the relationship between charge transfer, separation, and CO₂ reduction activity through *in-situ* transient absorption spectroscopy. However, the discrepancy in timescales between the transient absorption test and the catalytic reaction only allows us to establish an indirect connection between charge transfer and photocatalytic performance. Therefore, we employed *in situ* TRPL to investigate ultrafast interfacial electron transfer-induced carrier long lifetimes, serving as a bridge between transient absorption spectroscopy and CO₂ reduction performance.

Time-resolved fluorescence spectroscopy (TRPL) was conducted to investigate the long lifetime of all the photocatalysts. As presented in Supplementary Fig. 14, under an Ar atmosphere, the IN10 hybrid exhibits a longer average lifetime (τ_a) as respect to pristine In₂O₃ and Nb₂O₅ at an emission wavelength of 470 nm, where the fluorescence signals originate from both In₂O₃ and Nb₂O₅. Following the proposed S-scheme mechanism within the In₂O₃/Nb₂O₅ heterojunction, photoelectrons in the In₂O₃ CB migrate to the Nb₂O₅ VB and recombine with its photoholes, resulting in the accumulation of powerful electrons in the Nb₂O₅ CB and holes in the In₂O₃ VB, thereby prolonging the charge carrier lifetimes. Notably, the physically-mixed composite of In₂O₃ and Nb₂O₅ displays a shorter τ_a than the In₂O₃/Nb₂O₅ heterojunction, highlighting the significance of ultrafast interfacial electron transfer in extending carrier lifetimes. Furthermore, *in situ* TRPL was employed to explore the relationship between ultrafast charge transfer-induced carrier lifetimes and photocatalytic performance. The τ_a of IN10 in a CO₂ atmosphere is notably shorter compared to that in an Ar atmosphere (Supplementary Fig. 15a), suggesting that a substantial portion of photogenerated electrons in the Nb₂O₅ CB is involved in photoreactions with CO₂ molecules, consequently leaving fewer charge carriers available for recombination. Bare In₂O₃, Nb₂O₅, as well as their physically-mixed composite (Supplementary Fig. 15b-d), reveal almost identical decay curves in both CO₂ and Ar atmospheres, indicating poor photoreactions within these systems. Overall, the observed ultrafast interfacial charge transfer between In₂O₃ and Nb₂O₅ within the S-scheme heterojunction plays a triple role: preventing the recombination of self-carriers, efficiently separating powerful photoelectrons and photoholes, and extending their long-lifetimes. This is anticipated to significantly enhance the photocatalytic performance towards CO₂ reduction.

The related results and discussion have been included in the revised manuscript.

2. Material structure analysis

What are the specific structural differences between this material and existing catalysts, and how do we distinguish the atomic-level mixing described in this article? Judging from the EDS mapping characterization results in Figure 1 of the manuscript, the two are mixed together, which means that the two may form a metal oxide mixture rather than a heterojunction. This also explains why the difference in the proportion of added precursors significantly impacts the reaction. Is the difference in contrast seen in the TEM caused by the different degrees of mixing?

Response: (1) We believe it is difficult to form a metal oxide mixture or solid solution between Nb₂O₅ and In₂O₃ at 600 °C due to disparities in their metal ionic radii, valence states, and crystalline structures. Firstly, the smaller radii of Nb ions (0.64 Å) compared to In ions (0.80 Å) may induce lattice distortions upon attempted substitution, thereby hindering the formation of a metal oxide

mixture or solid solution. Secondly, the disparate crystalline structures of Nb₂O₅ and In₂O₃ (orthorhombic vs. cubic) pose another challenge to the metal oxide mixture formation. Thirdly, the +5-valence state of Nb (*e.g.*, Nb₂O₅) contrasts with the +3-valence state of In (*e.g.*, In₂O₃), potentially resulting in structural instability or difficulty in achieving charge balance within the solid solution.

(2) The specific structural difference between the In₂O₃/Nb₂O₅ heterojunction and other catalysts lies in the maximum phase contact without any hindrance achieved through the “one-pot” preparation method, which provides an unimpeded transport route and promotes interfacial charge transfer between In₂O₃ and Nb₂O₅. As evidenced in Fig. 1d, the HRTEM images of the In₂O₃/Nb₂O₅ heterojunction (IN10) reveal discernible two-phase grain boundaries with lattice fringes corresponding to In₂O₃ and Nb₂O₅, respectively.

Additionally, we attempted to verify the formation of a heterojunction rather than a metal oxide mixture by scanning elemental mappings across a restricted range. However, the random distribution of In₂O₃ and Nb₂O₅ grains within the nanofibers, along with approximately 100 nm thickness of the fibers, presented challenges in achieving an optimal elemental mappings for the two-phase particles.

Reviewer #4:

After checking the revised manuscript and the comments, I think the authors have made reasonable responses to the proposed comments and proper revision to the original one, thus it is suitable to be publishable in the currently-revised manuscript.

Response: Thank you very much.

REVIEWER COMMENTS

Reviewer #3 (Remarks to the Author):

The revised manuscript is now more thorough, which meets the publication requirements for Nature Communication. Finally, there are some suggestions to make the manuscript's evidence more convincing.

1. Evidence for reaction sites is lacking.

Regarding the reaction, it is recommended to supplement the blank control experiment, such as not adding CO₂, catalyst and hydrogen source, etc., to prove the reliability of the reaction results.

2. Evidence for material structure analysis

Regarding the material structure, it is recommended that the EDS map of the local structure in Figure 1e of the manuscript, with dimensions similar to that in Figure 1d, be supplemented to directly prove its reliability directly.

3. Others.

On page S22, part of the content is blocked, so it is recommended that Supplementary Figure 21 be redesigned.

Response to Reviewer' comments

Reviewer #3:

The revised manuscript is now more thorough, which meets the publication requirements for *Nature Communication*. Finally, there are some suggestions to make the manuscript's evidence more convincing.

1. Evidence for reaction sites is lacking.

Regarding the reaction, it is recommended to supplement the blank control experiment, such as not adding CO₂, catalyst and hydrogen source, *etc.*, to prove the reliability of the reaction results.

Response: We have conducted blank control experiments as recommended, under conditions without catalyst, CO₂, H₂O, and light irradiation, respectively. The corresponding results have been included in the revised manuscript.

2. Evidence for material structure analysis

Regarding the material structure, it is recommended that the EDS map of the local structure in Figure 1e of the manuscript, with dimensions similar to that in Figure 1d, be supplemented to directly prove its reliability directly.

Response: We have conducted energy-dispersive X-ray (EDX) elemental mappings, specifically targeting a grain-to-grain contact area within the IN10 nanofibers. Upon scrutinizing the enlarged local structure, it is evident that the distribution of Nb and In elements exhibits a non-overlapping pattern along the interface of the two phases (Fig. 1e). Further EDX elemental mappings of IN10 (Supplementary Fig. 2) unambiguously confirm the presence of In, Nb, and O elements within the composite nanofibers, offering compelling evidence for the formation of In₂O₃/Nb₂O₅ heterojunctions.

The results and discussion have been updated in the revised manuscript.

3. Others.

On page S22, part of the content is blocked, so it is recommended that Supplementary Figure 21 be redesigned.

Response: We have revised Supplementary Fig. 23 as per your suggestion. Many thanks.

REVIEWERS' COMMENTS

Reviewer #3 (Remarks to the Author):

The revised manuscript addressed the concerns and is recommended being published in Nature Communications.

Response to Reviewer' comments

Reviewer #3:

The revised manuscript addressed the concerns and is recommended being published in Nature Communications.

Response: Many thanks.